# Regulated delivery controls *Drosophila* Hedgehog, Wingless, and Decapentaplegic signaling

**Ryo Hatori, Brent M Wood, Guilherme Oliveira Barbosa, Thomas B Kornberg\***

Cardiovascular Research Institute University of California, San Francisco, San Francisco, United States

**Abstract** Morphogen signaling proteins disperse across tissues to activate signal transduction in target cells. We investigated dispersion of Hedgehog (Hh), Wnt homolog Wingless (Wg), and Bone morphogenic protein homolog Decapentaplegic (Dpp) in the *Drosophila* wing imaginal disc. We discovered that delivery of Hh, Wg, and Dpp to their respective targets is regulated. We found that <5% of Hh and <25% of Wg are taken up by disc cells and activate signaling. The amount of morphogen that is taken up and initiates signaling did not change when the level of morphogen expression was varied between 50 and 200% (Hh) or 50 and 350% (Wg). Similar properties were observed for Dpp. We analyzed an area of 150 μm×150 μm that includes Hh-responding cells of the disc as well as overlying tracheal cells and myoblasts that are also activated by disc-produced Hh. We found that the extent of signaling in the disc was unaffected by the presence or absence of the tracheal and myoblast cells, suggesting that the mechanism that disperses Hh specifies its destinations to particular cells, and that target cells do not take up Hh from a common pool.

## Introduction

Signaling by morphogen proteins controls many aspects of development, homeostasis, and disease (*Garcia et al., 2018*; *Tabata and Takei, 2004*; *Taipale and Beachy, 2001*). These signaling proteins are released from cells that produce them, and they distribute across the tissues they target, forming concentration gradients that induce signal transduction and activate gene expression in a concentration-dependent manner. The importance of regulation by morphogen gradients to growth, cell fate, and patterning underlies the imperative to understand how morphogens disperse across tissues.

For more than a century, it was assumed that morphogens spread across tissues by passive diffusion in extracellular space (either 'free' or 'restricted'), and both experimental observations and theoretical modeling have been offered in support (*Rogers and Schier, 2011*). Spreading morphogen proteins have been proposed to exist in various forms, including as multimeric complexes or encapsulated in lipoprotein particles, exosomes, or micelles (*Christian, 2012*). Implicit in these models are the ideas that signaling is proportional to amounts of signaling proteins produced by designated groups of cells, and that release creates an extracellular pool of signaling protein that distributes in extracellular fluid in ways that are dependent on interactions with substances that are encountered or until they are removed from the pool by degradation or by receptor-mediated absorption. The pool is assumed to be formed by constitutive release from producing cells.

An alternative mechanism of dispersion is direct exchange at cell-cell contacts, and the contrasts with diffusion-based modes of dissemination have been reviewed extensively (*Gradilla and Guerrero, 2013*; *Kornberg, 2017*; *Kornberg, 2016*; *Roy and Kornberg, 2015*). Specialized filopodia called cytonemes are conduits that transport and transfer signaling proteins to target cells at synaptic contacts (*González-Méndez et al., 2020*; *González-Méndez et al., 2017*; *Kornberg, 2016*). Evidence for the essential role of cytonemes in morphogen signaling includes many identified genetic

**\*For correspondence:**
tkornberg@ucsf.edu

**Competing interests:** The authors declare that no competing interests exist.

conditions that impair cytonemes, reduce cytoneme contacts, and compromise signaling. Although their fine structure is not fully understood, cytonemes contain actin filaments, ribosomes, and proteins that confer voltage sensitivity, calcium dependence, and glutamatergic activity (*Huang et al., 2019*; *Junyent et al., 2020*; *Wood et al., 2021*). These features, as well as the close apposition of pre- and postsynaptic membranes at cytoneme synapses, are also characteristic of neuronal glutamatergic synapses. Neuronal synapses have another key feature—signaling is titrated by frequency and quantity of neurotransmitter release from synaptic vesicles, and by efficiency of neurotransmitter clearance from the synaptic gap (*Blakely and Edwards, 2012*). It is not known if cytoneme synapses also store morphogen signaling proteins in synaptic vesicles and if morphogen signaling at cytoneme synapses is also dependent on regulated release and uptake.

Hedgehog (Hh), Wnt homolog Wingless (Wg), and Bone morphogenic protein homolog Decapentaplegic (Dpp) are evolutionarily conserved morphogen signaling proteins that have been implicated in organogenesis and stem cell maintenance, and their misregulation in mammals has been linked to inherited diseases and cancers (*Briscoe and Thérond, 2013*; *Morikawa et al., 2016*; *Nusse and Clevers, 2017*). In the columnar cells of the *Drosophila* wing imaginal disc, Hh is expressed specifically and uniformly by posterior (P) compartment cells (*Figure 1A*). In the wing blade primordium of the wing disc, Hh released by P compartment cells is taken up by anterior (A) compartment cells within 30 µm (10 cells) of the anterior/posterior (A/P) compartment border. Transfers of Hh from the P to A compartment cells are cytoneme-dependent (*Bischoff et al., 2013*; *Chen et al., 2017*). Hh in the A compartment distributes to form a concentration gradient that induces signal transduction and activates expression of target genes in partially overlapping stripes (*Callejo et al., 2011*; *Chen et al., 2017*). These domains of expression reflect graded responses to Hh, from highest and 'short-range' (*engrailed* [*en*], *patched* [*ptc*], and *dpp*) to lowest and 'long-range' (*cubitus interruptus* [*ci*]). The spatial relationships of these domains are reproducible, with single-cell resolution.

In the wing blade primordium, cells that express *dpp* form a stripe of 6–8 cells adjacent to the A/P compartment border (*Teleman and Cohen, 2000*). *wg* is expressed in a two cell-wide stripe that is orthogonal to the Dpp stripe and straddles the dorsal/ventral (D/V) compartment border (*Neumann and Cohen, 1997*). Both Dpp and Wg disperse to form concentration gradients on both sides of their respective stripes of expressing cells. In the wing disc, transport of Dpp is cytoneme-mediated (*Huang and Kornberg, 2015*; *Roy et al., 2014*) and although the role of cytonemes in Wg dispersion have not been investigated, Wnt signaling in zebrafish is cytoneme-mediated (*Stanganello et al., 2015*; *Stanganello and Scholpp, 2016*).

Here, we asked whether the distributions of Hh, Wg, and Dpp in cells of the wing disc are regulated in ways that might be analogous to distributions of glutamate at chemical neuronal synapses. More than 99% of glutamate in the brain is stored in presynaptic terminals and is active only after release in titrated amounts, but it is not known whether uptake of signaling proteins at cytoneme synapses is regulated. To investigate this question, we asked if the Hh, Wg, and Dpp distributions are dependent and proportional to the amount produced. We also asked if the three target tissues that respond to disc-produced Hh take up Hh from a common pool. Our data show that the delivery of signaling proteins to target cells is regulated with respect to both amount and destination.

## Results

### Relationship between Hh production and Hh signaling in the wing disc

Neurotransmitters that are made, packaged, and stored in presynaptic compartments are functionally inert, their precisely controlled release and delivery for juxtacrine activation a signature property of synaptic signaling. In order to investigate whether the release of Hh might be regulated at cytoneme synapses, we analyzed Hh signaling in genotypes that express different amounts of Hh. We tested whether amounts of Hh and Hh signaling in recipient cells are proportional to Hh production, as might be expected of constitutive, unregulated release by producing cells, or if they are independent of production as might be expected of regulated release and delivery.

We first monitored Hh signaling in wing discs with genotypes that vary the number of wildtype (WT) *hh* genes and *hh* transgenes (*Figure 1B*). The *hh* transgenes were (1) BAC plasmids containing the WT *hh* transcription unit in a genomic fragment of 40 kb (HS) or 101 kb (HL), or HS-GFP, the 40

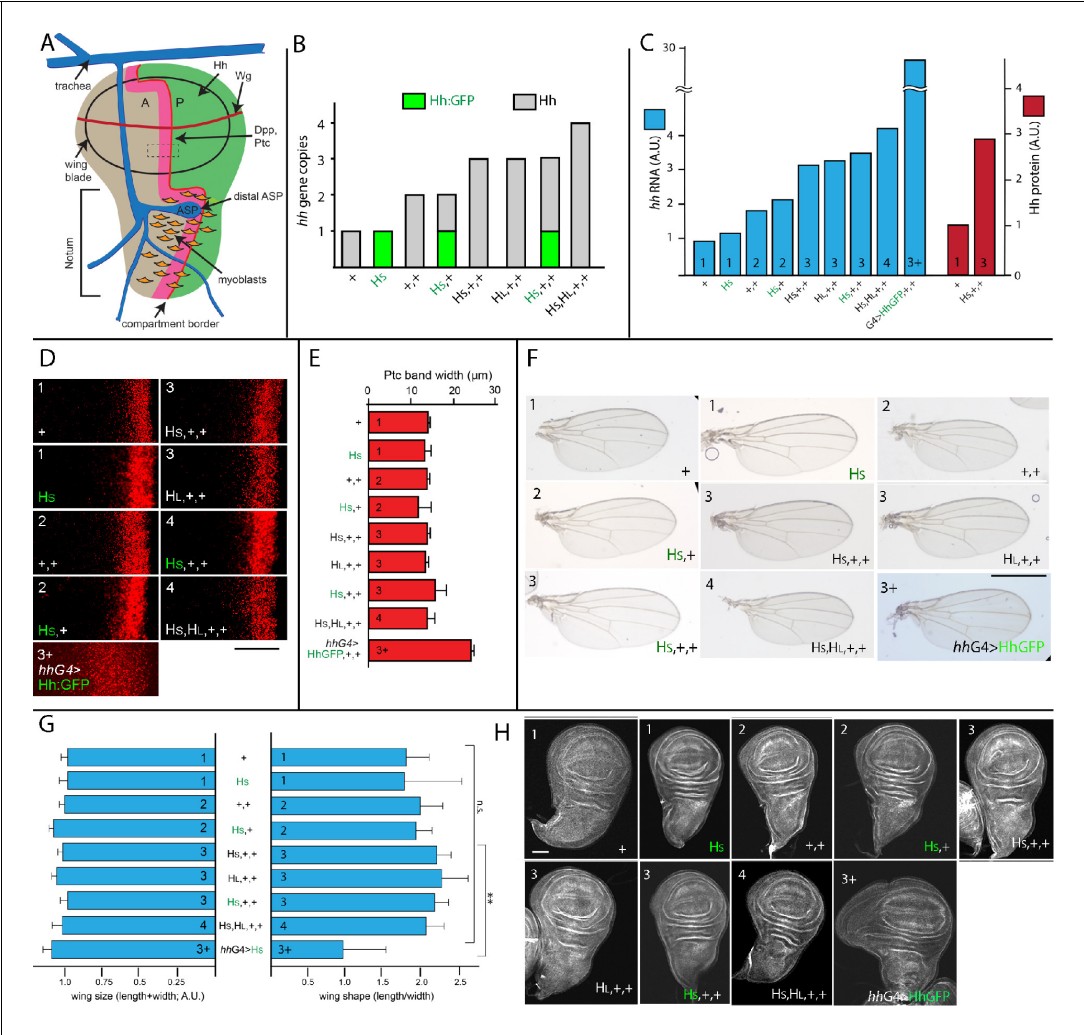

**Figure 1.** Signal transduction is constant in conditions that vary amounts of Hh production. (A) Schematic of the wing disc and ASP indicating A and P compartments, ASP and trachea (blue), myoblasts (orange), and domains of expression for Hh (green), Dpp and Ptc (pink), and Wg (red). Rectangle (dashed lines, 20 μm×20 μm) indicates region that was imaged at high magnification in (D) and in *Figure 2A,D*. (B) Bar graph showing the number of *hh* genes in genotypes with different combinations of WT *hh* and *hh* BAC transgenes; gray and green bars represent genes encoding Hh and Hh:GFP, respectively. (C) Bar graph showing the amount of Hh RNA (blue) in wing discs and Hh protein (red) in wing disc P compartments, measured by qPCR and α-Hh antibody staining, respectively, with genotypes indicated and number of *hh* genes indicated by numbers in the bars; values normalized to the amount of *hh* RNA and Hh protein in genotype with 1 copy of WT *hh* (+). (D) Optical sections showing α-Ptc antibody staining in region indicated in (A) by rectangle for indicated genotypes. Scale bar: 20 μm. (E) Bar graph of widths of antibody stained Ptc domains in (D), manually measured from maximum intensity projections of optical sections spanning 10 μm from the most apical side of the wing pouch cells. No statistically significant differences for 1–4 gene copies (p>0.05), n=6–8 for each genotype. (F) Adult wings for indicated genotypes. Scale bar: 100 μm. (G) Bar graph showing the measured wing size (left) and wing shape (right); no statistically significant differences (p>0.05), n=12–18 for each genotype. (H) Wing discs for each indicated genotype. Error bars in (D, G) indicate standard deviation (SD). Scale bar: 100 μm. Genotypes: +- (WT *hh* gene); H_S (*40 k Hh BAC*); H_L- (*100 k Hh BAC*); H_S (*Hh:GFP 40 k BAC*); + (*hh^{AC}/+*); H_S, (*Hh:GFP 40 k BAC; hh^{AC}/hh^{AC}*); H_S,+ (*Hh:GFP 40 k BAC; hh^{AC}/+*); H_S,+,+ (*Hh 40 k BAC; +/+*); H_L,+,+ (*Hh 100 k BAC; +/+*); H_S,+,+ (*Hh:GFP 40 k BAC; +/+*); H_S,H_L,+,+ (*Hh 40 k BAC / Hh 100 k BAC +/+*); hhG4>HhGFP (*hh*Gal4 UAS-HhGFP, +/+).

The online version of this article includes the following source data and figure supplement(s) for figure 1:

**Source data 1.** Amount of Hh mRNA.
**Source data 2.** Ptc band width.
**Figure supplement 1.** Apical/basal distributions of Ptc in *hh* gene copy number genotypes.
**Figure supplement 2.** Intensity profiles of Ptc expression in *hh* gene copy number genotypes.
**Figure supplement 3.** Knot expression in *hh* gene copy number genotypes.
**Figure supplement 3—source data 1.** Kn band width.
**Figure supplement 4.** Colocalization of Hh and Rab7 in the wing imaginal disc.

kb genomic fragment into which GFP has been recombined in frame (Hh:GFP) (*Chen et al., 2017*); and (2) a Hh:GFP construct expressed under GAL4 control. Flies without a functional *hh* gene die as embryos, but haploid flies with one BAC transgene (encoding Hh [HS, HL, or HS-GFP]) had normal appearance and wing discs had normal morphology (*Chen et al., 2017*). Hh:GFP encoded by this transgene is therefore presumed to be a functional surrogate for the normal, WT protein.

To investigate how the production of *hh* RNA and Hh protein in the wing disc scale with gene dosage, we measured amounts of *hh* RNA by qPCR and amounts of Hh protein by monitoring α-Hh antibody staining (*Figure 1C*). Genotypes with 1 (1× WT), 2 (2× WT), 3 (2× WT, 1× BAC), or 4 (2× WT, 2× BAC) *hh* genes had amounts of *hh* RNA in wing discs which increased with gene dosage and scaled linearly. Hh protein in the P compartment (Hh-producing) cells of the wing blade primordium increased 2.9× in genotypes with 3 *hh* genes versus 1. These results show that in this range of gene copy number, both *hh* RNA and Hh protein are produced in direct proportion to gene dosage. We also measured *hh* RNA in wing discs with 2 WT *hh* genes and transgenes containing *hh-Gal4* and *UAS-hh:GFP*, and observed that the amount increased approximately 15× over WT.

To determine if different amounts of Hh expression change signaling, growth, and patterning, we examined several parameters which are sensitive to Hh signaling: expression of a gene targets of Hh signal transduction, and size and shape of the wing and wing disc. Expression of the *ptc* gene in the wing disc is upregulated by Hh signaling in a band of cells at the A/P compartment border. The width of this band decreases under conditions of reduced Hh signal transduction (*Molnar et al., 2011*), and increases under conditions in which Hh signaling is elevated (*Cheng et al., 2012*; *Wang and Holmgren, 1999*). As shown in *Figure 1D,E*, the size of the Ptc band did not change in discs with 1, 2, 3, or 4 *hh* gene copies, despite the differences in *hh* RNA and Hh protein amounts (*Figure 1D,E*, *Figure 1—figure supplement 1* and *2*). In contrast, Hh overexpression driven by *hh-Gal4* changed both the size and shape of the Ptc band. Expression of Knot (Kn), which like Ptc expression is regulated by Hh signaling (*Vervoort et al., 1999*), did not change in genotypes with 1–4 *hh* gene copies (*Figure 1—figure supplement 3*).

We monitored wing size in the different genotypes by measuring the length (A/P border length from wing tip to wing base) and width (length of a line extending the posterior crossvein from anterior to posterior margin). These wing dimensions were not dependent on *hh* gene copy or expression (*Figure 1F,G*). We monitored wing shape by comparing the length/width ratios of the different genotypes, and determined that only the *hh-Gal4*-driven (~15×) overexpression changed it significantly (*Figure 1G*). L3 wing discs were also similar in appearance except for the *hhGal4* genotype, which had an outgrowth anterior to the wing blade primordium (*Figure 1H*). The only apparent change we noted to the adult morphology was the decrease in length/width ratio.

These findings indicate that the processes of production, maturation, secretion, transport, and uptake that disperses Hh in the wing disc generates a distribution in targets cells that is insensitive to changes in Hh amounts that 1–4 *hh* gene copies generate. The sensitivity to an excess of ~15-fold normal Hh generated by *hh-Gal4* reveals that the capacity of the process which buffers against changes in amounts of Hh production is limited. We do not know which step or steps in the process might be overwhelmed by the ~15× excess—if, for instance, Hh released in this overexpression condition is processed and modified normally, or if overexpressed Hh exits by the normal route. The important point is that there is a system that can compensate for different amounts of production. Previous studies characterized the robustness of Hh signaling to variations in either production or response, assuming that release from producing cells is constitutive and that robustness is solely an attribute of the signal transduction process (*Li et al., 2018*; *Zhang et al., 2020*). We consider two possible alternatives.

If the amount of Hh taken up by the recipient, target cells are proportional to production, each recipient cell might scale the outputs of Hh signal transduction relative to its neighbors. This mechanism might adjust relative responses independently of absolute amounts, determining growth and pattern by the slope of the concentration gradient across a field of cells. This type of mechanism was proposed for the morphogen gradient of Dpp in order to model the effects of mosaic ectopic activation induced by the expression of a constitutively active Dpp receptor (*Rogulja and Irvine, 2005*). Alternatively, the amount of Hh transferred from producing to responding cells might be regulated independently of the amount produced. To distinguish between these mechanisms, we quantified Hh in recipient cells of the wing blade primordium.

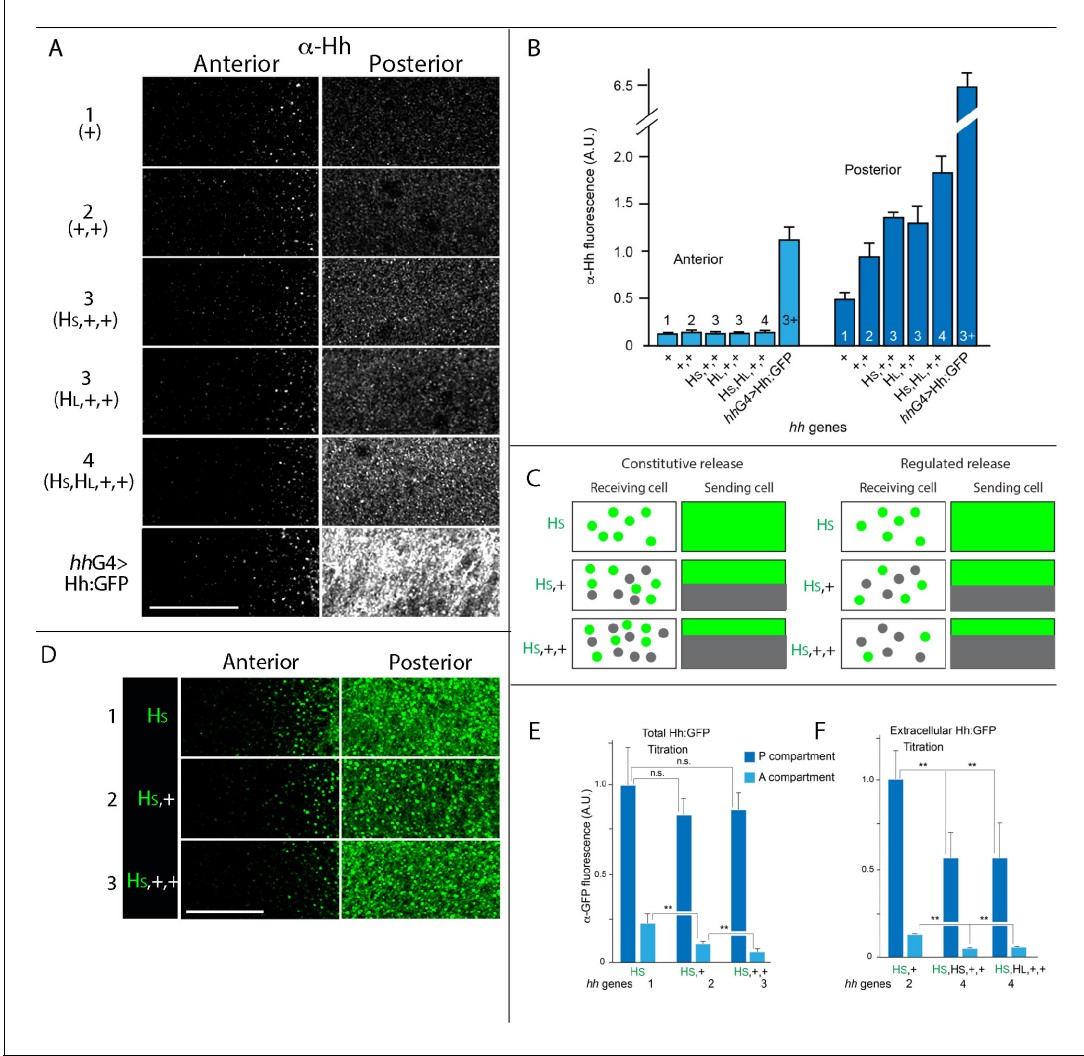

**Figure 2.** Hh delivery is constant in conditions that vary amounts of Hh production. (A) α-Hh antibody staining in regions indicated in *Figure 1A* for indicated genotypes. (B) Bar graph showing the intensity of α-Hh antibody staining in A and P compartments of wing blades for indicated genotypes. No statistically significant differences for A compartment (p>0.05) except for *hhGal4*>Hh:GFP; for P compartment, staining was statistically different for genotypes with different numbers of *hh* genes (1, 2, 3, and 4, *hhGal4*; Student's t-test, p<0.05), but not between equivalent numbers of genes (HS,+, + and HL,+,+; p>0.05), n=5–7 discs for each genotype. (C) Schematic portraying the predicted differences between constitutive release and regulated release for different genotypes, Hh and Hh:GFP indicated by gray and green dots, respectively. (D) Images of α-GFP antibody staining in regions indicated in *Figure 1A* for indicated genotypes. (E, F) Bar graphs showing wing discs with indicated genotypes stained with α-GFP antibody using standard fixation (E) or extracellular staining protocol (F). **-p<0.005, n.s.-p>0.05; n=4–6 for each genotype. Abbreviations as in *Figure 1*. Scale bar: 20 μm.

The online version of this article includes the following source data and figure supplement(s) for figure 2:

**Source data 1.** Amount of Hh.

**Source data 2.** Amount of Hh:GFP.

**Source data 3.** Amount of extracellular Hh:GFP.

**Figure supplement 1.** Apical/basal distributions of Hh in *hh* gene copy number genotypes.

**Figure supplement 2.** Distributions of extracellular Hh in *hh* gene copy number genotypes.

**Figure supplement 3.** Dextran uptake in *hh* gene copy number genotypes.

**Figure supplement 3—source data 1.** Amount of punctae with Hh:GFP and Dex.

Quantification of Hh detected by antibody staining in the entire A and P compartments of the wing primordium of normal discs (2× WT) showed that approximately 5.2% was in the A compartment (n=7; standard deviation [SD]=1.2%). We next monitored Hh with α-Hh antibody in the small rectangular region composed of equal portions of the A and P compartments (*Figure 1A*) in

genotypes with 1, 2, 3, or 4 *hh* genes. As expected, Hh amounts were proportional to gene dosage in the P compartment portion of the rectangular region. The cells in this region produce Hh and this result is consistent with the analysis of the entire P compartment (described above) and the finding that Hh RNA and protein scale with gene dosage. Analysis of the anterior portion of the rectangular region revealed that Hh amounts were not detectably different in genotypes with 1, 2, 3, or 4 *hh* genes (*Figure 2A,B*, *Figure 2—figure supplement 1*). In sum, these results are consistent with the idea that most Hh produced in the P compartment is not taken up or retained by A compartment cells, and that Hh uptake is not linked directly to production. We also examined the intracellular localization of Hh to determine if it colocalized with Rab7, which concentrates in late endosomes. Association with Rab7 would suggest that a fraction of Hh produced in P compartment cells is destined for turnover, not release (*Figure 1—figure supplement 4*).

To characterize the relationship between Hh production and delivery further, we used an α-GFP antibody to analyze genotypes with one Hh:GFP-encoding BAC transgene (BHS-GFP) together with either zero, one, or two (untagged) WT *hh* genes (Total genes: 1: BH-GFP; −/−; 2: BHS-GFP; +/−; and 3: BHS-GFP; +/+) (*Figure 1B*). As depicted in *Figure 2C*, α-GFP antibody staining of GFP-tagged Hh that is titrated with different amounts of untagged Hh distinguishes between constitutive and regulated delivery in these genotypes. If delivery of Hh to the A compartment is proportional to gene dosage and not regulated, Hh:GFP amounts in the A compartment are expected to be unaffected by co-production of untagged Hh, and Hh:GFP remains constant as gene dosage and production increases. However, if delivery is regulated, the fraction of Hh:GFP in the A compartment is expected to decrease as the fraction of untagged Hh increases in proportion to total gene copy and production.

Analysis of wing discs with one Hh:GFP BAC (BHS-GFP) and zero, one, or two WT *hh* genes shows that Hh:GFP in the producing cells of the P compartment was not diminished by the presence of *hh* genes that encode untagged Hh (*Figure 2D,E*). This is consistent with the idea that the production of both Hh RNA and protein are proportional to gene copy. In contrast, Hh:GFP amounts in the Hh-receiving cells of the A compartment decreased in proportion to number of *hh* genes that encode untagged Hh. This shows that Hh:GFP was diluted by the presence of untagged Hh, a result which is consistent with the amounts of Hh we detected in the A compartment with α-Hh antibody in genotypes with 1, 2, 3, or 4 *hh* genes (*Figure 2A,B*). We conclude that the amount of Hh in the A compartment was constant and did not scale with production.

## Gene dosage dependence of extracellular Hh

We applied an extracellular staining protocol that detects antibodies bound to preparations of non-permeabilized and unfixed cells (*Strigini and Cohen, 2000*). We stained wing discs with a high titer α-GFP antibody that detected BAC-encoded Hh:GFP, and analyzed genotypes with 1 or 3 genes that encode untagged Hh. For reference, total GFP fluorescence was determined in discs processed with our standard immunohistochemistry protocol that includes detergent permeabilization and formaldehyde fixation. GFP fluorescence increased in P compartment cells and decreased in A compartment cells in proportion to *hh* gene copy (*Figure 2E*). The extracellular staining protocol detected basolateral GFP fluorescence (*Figure 2—figure supplement 2*), consistent with the observations of *Callejo et al., 2011* who reported that Hh moves to the basolateral compartment prior to localization to basolateral cytonemes and export (*Callejo et al., 2011*). Hh:GFP detected by the extracellular staining protocol was less than the total, as the fluorescence was visible only in the most basal optical sections and required higher laser power and gain settings (see Materials and methods), but direct quantitative comparisons are not possible because of the different protocols. Extracellular staining decreased in both P and A compartments in the presence of untagged Hh (*Figure 2F*), indicating that Hh exposed on the exterior of P compartment cells is gated. Although the results suggest that Hh taken up by A compartment cells represents the population of externalized Hh present on the surface of P compartment cells, they do not reveal whether the release of externalized Hh is regulated.

## Dpp production and signaling in the wing disc

To investigate whether regulated delivery is also a feature of Dpp signaling, we monitored Dpp signaling and dispersion in genotypes with different numbers of *dpp* genes. We created a Dpp-

encoding BAC transgene (BD) that rescues *dpp* haploinsufficiency: animals with one WT *dpp* and one BD (+/*dpp*$^{H46}$; +/BD) allele are viable and their wing size is comparable to WT flies (**Figure 3A, B**), indicating that the Dpp BAC is a functional substitute for a WT *dpp* gene. To monitor different amounts of *dpp* expression, we compared wing discs with two or four *dpp* genes (two genes: +/+; four genes: +/+; BD/BD). First, to examine proportionality between *dpp* gene copy and Dpp protein, we stained wing discs with antibody that recognizes the prodomain of unprocessed Dpp (**Akiyama and Gibson, 2015**; **Panganiban et al., 1990**). Staining in cells that produce Dpp was

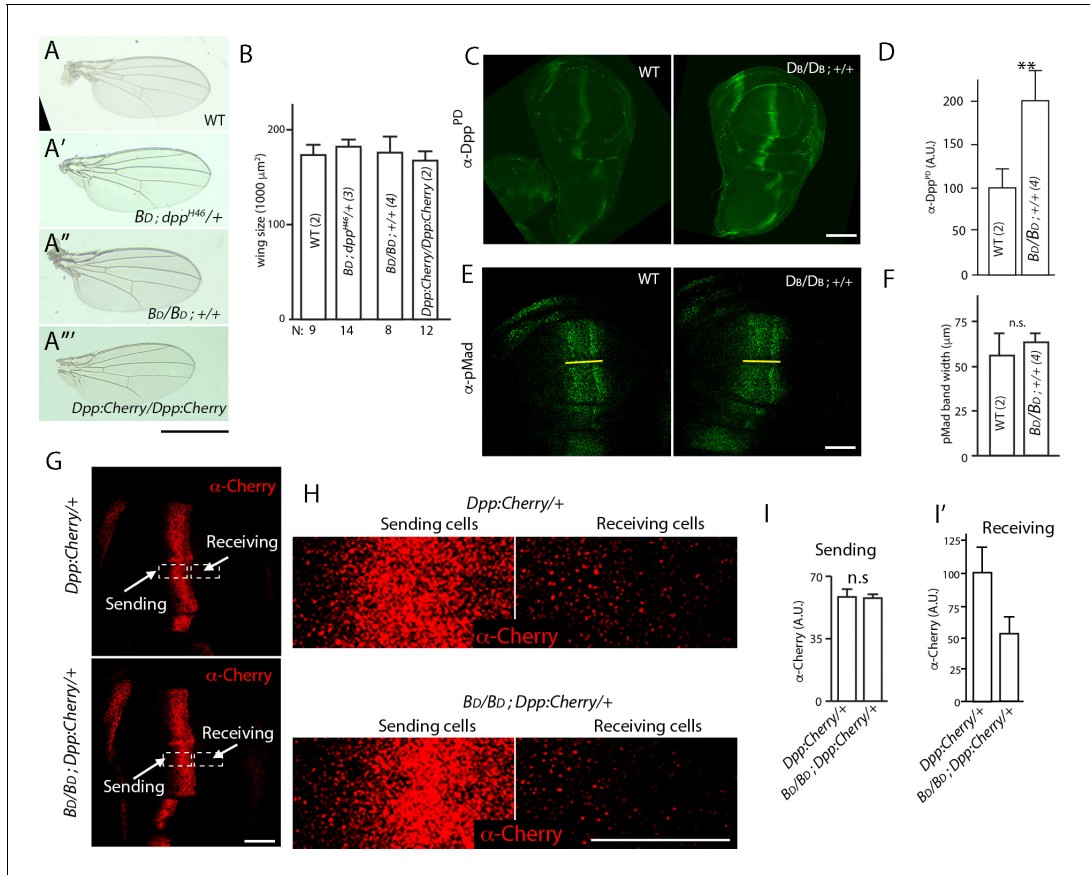

**Figure 3.** Dpp delivery and signal transduction are constant in conditions that vary amounts of Dpp production. (A–A''') Adult wings for indicated genotypes. Scale bar: 100 µm. (B) Bar graph showing size of adult wings for genotypes in (A–A'''); error bars indicate SD, N indicates number of wings analyzed; no statistically significant differences indicated by Student's t-test (p>0.05). (C) Wing discs with two (WT) and four *dpp* genes: WT (+/+); four (BD/BD; +/+) stained with α-Dpp prodomain (α-DppPD) antibody; scale bar: 100 µm. (D) Bar graph quantifying α-DppPD antibody staining for wing discs with indicated genotypes, n=7 (two genes) and 8 (four genes). Difference is statistically significant (Student's t-test [p<0.005]). (E) Images of wing discs with indicated genotypes stained with α-pMAD antibody; yellow line marks the width of pMAD band; scale bar: 50 µm. (F) Bar graph quantifying α-pMAD antibody staining for wing discs with indicated genotypes; n=7 (two genes) and 3 (four genes). No statistically significant differences indicated by Student's t-test (p>0.05). (G) Wing discs with (upper panel) one untagged Dpp (+) and one Dpp:Cherry encoding gene, or (lower panel) three untagged Dpp and one Dpp:Cherry encoding gene stained with α-Cherry antibody; scale bar: 50 µm. (H) High magnification images of boxed regions in (G); scale bar: 25 µm. (I, I') Bar graphs quantifying α-Cherry antibody staining in sending (I) and receiving regions (I') for indicated genotypes; error bars indicate SD; (I) no statistically significant differences indicated by Student's t-test (p>0.05); n=7 for each genotype. (I') Difference is statistically significant (p<0.05). BD, Dpp-encoding BAC transgene; Dpp:Cherry, Dpp:Cherry knock-in allele.

The online version of this article includes the following source data and figure supplement(s) for figure 3:

**Source data 1.** Wing size.
**Source data 2.** Amount of Dpp.
**Source data 3.** pMad band width.
**Source data 4.** Amount of Dpp:Cherry (Sending).
**Figure supplement 1.** Salm expression in *dpp* gene copy number genotypes.
**Figure supplement 1—source data 1.** Spalt band width.

approximately double in the four copy genotype compared to the two copy genotype (*Figure 3C, D*). Second, we examined wing size, which is sensitive to different amounts of Dpp signaling. Mutant conditions that decrease Dpp signal transduction reduce wing disc growth and mutant conditions that elevate signal transduction cause overgrowth (*Capdevila and Guerrero, 1994*; *Spencer et al., 1982*). We found that wing size did not differ between genotypes with two or four *dpp* genes (*Figure 3A,B*). Third, we asked if signal transduction increases with gene dosage and Dpp production. α-pMAD staining, a readout of Dpp signaling, forms a band that coincides with and straddles *dpp* expressing cells in WT discs. The width of the pMAD-staining band was not different in wing discs with two or four gene copies (*Figure 3E,F*), indicating that increased Dpp production does not increase signal transduction. Similarly, expression of Spalt (Salm), which is regulated by Dpp signaling in the wing disc (*de Celis et al., 1996*), did not change in genotypes with two or four *dpp* gene copies (*Figure 3—figure supplement 1*). In sum, these results show that Dpp signaling in the wing disc is insensitive to increased levels of Dpp production.

To monitor Dpp distributions, we examined discs stained with α-Dpp antibody to compare genotypes with one Dpp:Cherry knock-in allele (generated by CRISPR-mediated recombination) and either one or three *dpp* genes that encode untagged protein (*Figure 3G*). The experimental setup and rationale are similar to the analysis of Hh:GFP depicted in *Figure 2C*. Evaluation of the two genotypes showed that levels of Dpp:Cherry fluorescence in producing cells did not change with the production of untagged Dpp (*Figure 3H,I*), and that levels of Dpp:Cherry fluorescence in non-producing, receiving cells decreased in proportion to the number of genes that encode untagged Dpp (*Figure 3I'*). This finding, that the amount of Dpp:Cherry in target cells decreased as the ratio of tagged:untagged Dpp declined, is consistent with the idea that transmission of Dpp to targets is regulated.

## Wingless production and signaling in the wing disc

We investigated Wg dispersion by analyzing three genotypes with different numbers of functional *wg* genes: 1 ($wg^+/wg^-$), 2 ($wg^+/wg^+$), and $2^{+overexpression}$ (*wg-Gal4 UAS-Wg:GFP; $wg^+/wg^+$*), and monitoring expression and distribution of Wg, as well as the Wg gene targets *senseless* (*sens*) and *Distal-less* (*Dll*). α-Wg antibody staining showed that Wg production is proportional to gene copy number in discs with one and two *wg* genes, and that the *wg-Gal4* driver generated approximately seven times more Wg than a single endogenous gene (*Figure 4A,A',B*). Despite the differences in expression between these genotypes, the amount of Wg in the neighboring cells that received Wg was unchanged (*Figure 4A,A',B*). The fraction of Wg present in the neighboring cells relative to the total produced in the wing blade decreased with increasing functional gene dosage, from approximately 41% (one copy) to 24% (two copies), and 3.5% (seven functional equivalents). α-Sens antibody detects two narrow, parallel stripes of expression that are immediately adjacent to but do not overlap the Wg-expressing cells (*Nolo et al., 2000*), and the patterns of Sens expression were not detectably different in these genotypes (*Figure 4A'',C,C'*). α-Dll antibody detects a region of expression that overlaps Wg-expressing cells (*Zecca et al., 1996*), and the pattern of Dll expression also was not different in these genotypes (*Figure 4—figure supplement 1*). These findings indicate that Wg transmission to target cells is regulated.

We investigated the distribution of Wg using the extracellular staining protocol and detected low levels of basolateral Wg in Wg-producing (*Figure 4—figure supplement 1*). Similar to Hh (*Callejo et al., 2011*), Wg moves to the apical surface before relocating to the basolateral surface (*Yamazaki et al., 2016*), and the small fraction of basolateral, extracellular Wg suggests that insertion in the basolateral membrane is gated. To determine if this portion of the Wg population is sensitive to Wg production amounts, we examined discs that expressed Wg from one or two *wg* genes, or from two *wg* genes combined with *wgGal4* driven overexpression. We found that extracellular Wg of both producing and receiving cells increased with gene dosage and expression (*Figure 4D*). This protocol does not distinguish between protein on either the cell surface or associated with cytonemes, but these results suggest that delivery and release to receiving cells from the basolateral membrane is rate-limiting.

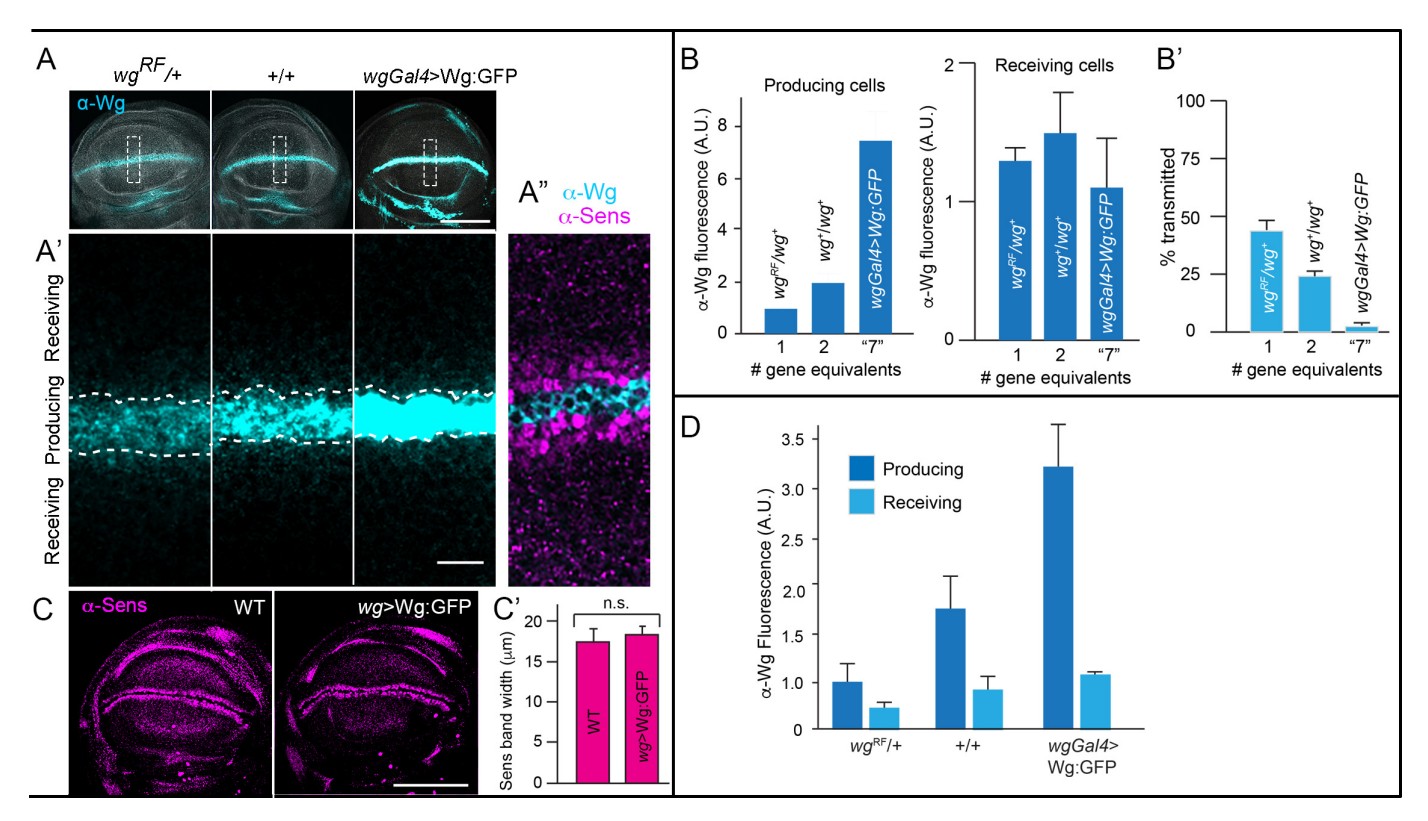

**Figure 4.** Wg signal transduction is constant in conditions that vary amounts of Wg production. (A–A') Wing blades for indicated genotypes stained with α-Wg antibody (cyan) and phalloidin (gray); one gene ($wg^−$/+), two genes (+/+), overexpression (wgGal4>UAS-wg; +/+). Scale bar: 100 µm. (A') higher magnification images of boxed regions (30 µm×90 µm) in (A), dashed white lines mark boundary between producing and receiving cells. Scale bar: 10 µm. (A'') Optical section of region similar to (A') stained with α-Wg (cyan) and α-Sens antibodies (magenta). (B) Bar graphs quantifying α-Wg staining for indicated genotypes; n=5–6 for each genotype. Values are normalized to the intensity of α-Wg staining for $wg^{RF}/wg^+$ (1 copy of wg gene). # gene equivalents indicate approximate Wg production functionality for each genotype. Difference in the producing cells is statistically significant (Student's t-test [p<0.005]), while difference in the receiving cells is not (Student's t-test [p>0.05]). (B') Bar graph quantifies the fraction of Wg in the receiving cell as % of total wing blade α-Wg antibody intensity in receiving cell. Statistical significance indicated by p<0.0005. (C) Wing blades with two WT genes or Wg overexpression (wgGal4>UAS-Wg:GFP; +/+) stained with α-Sens antibody. Scale bar: 100 µm. (C') Bar graph quantifies the width of α-Sens antibody stained band in maximum intensity projections of optical sections for entire apical-basal depth; 10 length measures were taken for each disc; no statistically significant differences (p>0.05); n=4 for each genotype. Genotypes: wg⁻/+ ($wg^{RF}$/+); +/+ (WT); wg-Gal4>wg (wg-Gal4; UAS-Wg: GFP / +/+).

The online version of this article includes the following source data and figure supplement(s) for figure 4:

**Source data 1.** Amounts of Wg.
**Source data 2.** Sens band width.
**Source data 3.** Amounts of extracellular Wg.
**Figure supplement 1.** Dll expression in *hh* gene copy number genotypes.

## Relationship between cytonemes and Hh production

Previous studies in several different contexts showed that the number of cytonemes correlates positively with signal transduction activity. Whereas cells with low signaling activity have few cytonemes, cells with higher levels of signaling have more (*Bischoff et al., 2013*; *Chen et al., 2017*; *Du et al., 2018*; *González-Méndez et al., 2017*; *Huang et al., 2019*; *Huang and Kornberg, 2016*; *Mattes et al., 2018*; *Roy et al., 2011*; *Roy et al., 2014*). This correlation also holds for mutant conditions that change cytoneme numbers or signaling: an example is overexpression of Ihog which increases cytoneme stability and Hh signaling in the wing disc (*González-Méndez et al., 2017*).

We first analyzed Hh signaling and cytoneme densities in the air sac primordium (ASP), a tracheal branch that is physically attached to the wing disc and which extends cytonemes to the disc that mediate the uptake of Hh, Dpp, and Bnl (*Chen et al., 2017*; *Du et al., 2018*; *Hatori and Kornberg,*

2020; *Roy et al., 2014*). To investigate the relationship between cytonemes and Hh production amounts, we monitored ASP cytonemes in genotypes with different numbers of *hh* genes.

We first asked if delivery of Hh to ASP cells is sensitive to amounts of Hh production by monitoring two conditions that are dependent on Hh signaling in the ASP: tissue morphology and expression of *engrailed* (*en*), which is a transcriptional target that is induced by Hh signaling in the wing blade and ASP (*Guillen et al., 1995*; *Hatori and Kornberg, 2020*). In WT, the ASP has a proximal narrow stalk and distal bulb (*Figure 1A*), and *en* expression is graded, with highest levels in the tip cells (*Figure 5A,B*). In mutant conditions with elevated Hh signaling (e.g., ectopic overexpression of Hh in the ASP), the stalk was absent and En expression extended to more proximal tracheal cells (i.e., the transverse connective; *Figure 1A*), whereas in mutant conditions with reduced Hh signaling (e.g., *smoothened* loss-of-function and Patched overexpression), the stalk was elongated and En expression was reduced. We found that in genotypes with 1–4 *hh* genes, neither ASP morphology nor extent of En expression changed (*Figure 5A–C*). In contrast, Hh overexpression driven by *hh-Gal4* reduced the stalk and increased the extent of En expression. These results show that the ASP is insensitive to the different amounts of Hh produced by 1–4 gene copies, and are consistent with the idea that Hh delivery is regulated. The sensitivity to *hh-Gal4* driven overexpression indicates that the capacity of the ASP system to buffer against different levels of expression is limited.

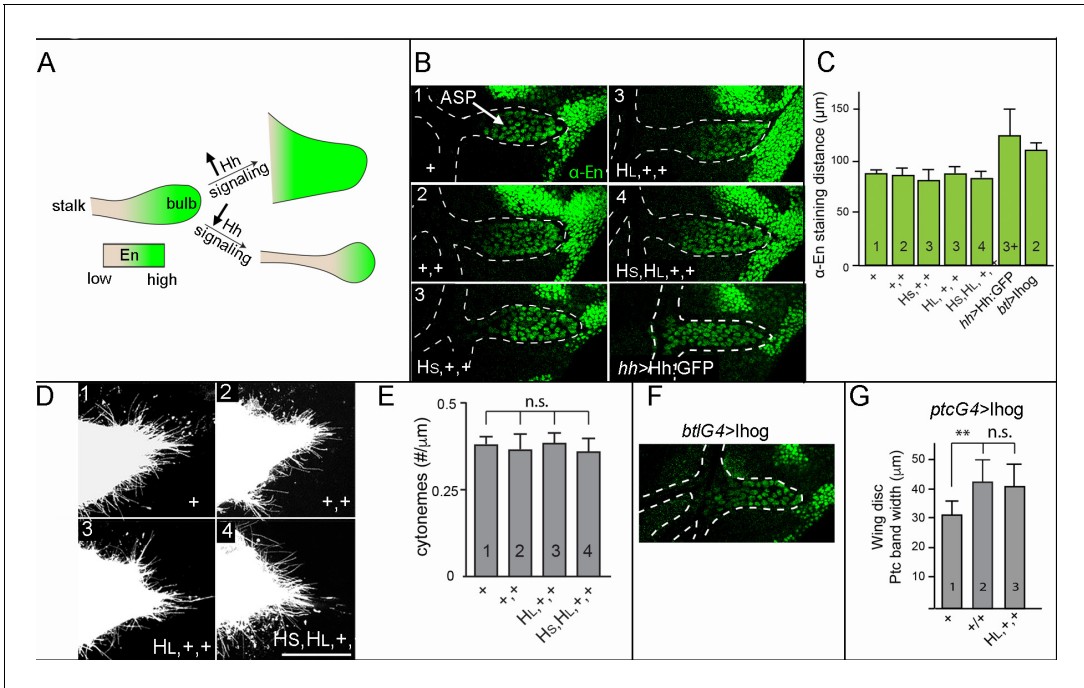

**Figure 5.** Neither signal transduction nor cytoneme number scales with Hh production in the ASP. (A) Schematic showing En expression (green) in WT ASP (left), ASP with high levels of Hh signal transduction and no stalk (top right), or ASP with low levels of Hh signal transduction and elongated stalk (bottom right). (B) α-En staining (green) of ASPs (bulb within white dashed line) for indicated genotypes (number of *hh* genes indicated in upper left). (C) Bar graph quantifying the distance of α-En antibody staining from the tip of the ASP toward the stalk for indicated genotypes (numbers of genes indicated in bars); no statistically significant differences indicated by p>0.05, n=5–7 for each genotype. (D) Cytonemes marked by the expression of *Cherry:CAAX* (*btl-lexA>lexO-Cherry:CAAX*) in the ASP for indicated genotypes (number of genes indicated in upper left). (E) Number of cytonemes for indicated genotypes (number of genes indicated in bars). Statistically significant differences indicated by p>0.05, n=5 for each genotype. (F) Ectopic overexpression of Ihog in the ASP reduced the stalk and increased extent of α-En staining (green). (G) Width of α-Ptc staining band in the wing discs ectopically overexpressing Ihog in the indicated genotypes; differences between 1 and 2 *hh* copies statistically significant (p<0.05). Abbreviations as in *Figure 1*.

The online version of this article includes the following source data for figure 5:

**Source data 1.** Cytoneme density.
**Source data 2.** Ptc band width.

We next investigated the relationship between Hh production and ASP cytonemes. We analyzed the number of cytonemes in genotypes with 1, 2, 3, and 4 *hh* genes by marking ASP cytonemes with membrane-tethered Cherry (*btl>CD8:Cherry*). Cytonemes that extend from the distal tip of the ASP take up Hh and contain Ptc (*Chen et al., 2017*), and in the experimental genotypes with 1–4 *hh* genes, the number of distal tip cytonemes did not change (*Figure 5D,E*). We conclude that the number of cytonemes and amount of cytoneme-mediated Hh uptake are insensitive to conditions that reduce or increase Hh production by a factor of 2 relative to WT.

We also investigated the effects of Ihog over-expression, which stabilizes cytonemes. The extent of En expression in the ASP increased under conditions of Ihog over-expression (*btlGal4 UAS-Ihog*), consistent with the idea that stabilized cytonemes increased Hh uptake and signaling (*Figure 5F*). In the wing disc, Ihog over-expression in the *ptc* domain at the A/P compartment border (*ptcGal4 UAS-Ihog*) increased both the width of the Ptc band (compare *Figure 1E*, *Figure 5G*) and sensitivity to amount of Hh production. In genotypes with 1, 2 or 3 *hh* genes, the width of the Ptc band in discs with two copies increased approximately 35% relative to discs with one (ANOVA and Tukey; p<0.05), but no significant difference between discs with two or three copies (*Figure 5G*). These results show both that genetic conditions that increase cytoneme numbers also increase Hh signaling of wing disc cells, and that cells that overexpress Ihog have a limited capacity to respond to increases in Hh.

## Expression of modulators of morphogen protein signaling

Morphogen signaling is a multi-step process that involves post-translational processes that prepare Hh, Dpp, and Wg in producing cells, feedback regulation in receiving cells, and extracellular proteins that influence activity. We investigated whether changes in the production of Hh, Dpp, or Wg affect the expression of genes that encode functions known to modulate signaling, because the expression of these genes might provide feedback regulation that compensates for changes in the amounts of proteins that are released or taken up. We might expect, for example, that the expression of a gene that provides negative feedback increases under conditions of increased signaling protein production. *Shifted* (*Shf*), for instance, encodes an extracellular factor that is required for the normal distribution of Hh, and Shf protein levels decrease in conditions of lowered signaling (*Glise et al., 2005*). We quantified *shf* expression in discs with one and four *hh* genes by quantifying shf transcripts with qPCR; no change in *shf* mRNA was detected in these genotypes (*Figure 6A,B*). This insensitivity to the tested changes in Hh amounts suggests that Shf does not control Hh release. *brinker* (*brk*), *pentagone* (*Pent; aka magu*), *short gastrulation* (*sog*), and *crossveinless-2* (*Cv-2*) negatively affect Dpp signaling. Brk is a transcriptional repressor of Dpp signal transduction whose expression is suppressed by Dpp. *Pent, Sog,* and *cv-2* encode extracellular proteins that bind Dpp and negatively affect spread and signaling (*Raftery and Umulis, 2012*; *Figure 6A*). Ectopic Dpp signaling suppresses *brk*, *pent*, and *sog* and upregulates *cv-2* expression (*Raftery and Umulis, 2012*; *Yu et al., 1996*). qPCR analysis detected no changes to expression of *brk, pent, sog,* or *cv-2* in genotypes with two or four *dpp* genes (*Figure 6B*). *Notum* expression is induced by Wg signaling and encodes an extracellular deacylase of Wg that inhibits Wg signaling (*Minami et al., 1999*), but its expression is not influenced by changes to Wg gene number (*Figure 6B*). In sum, these data do not support the idea that expression of known modulators of the Hh, Dpp, and Wg pathways compensate for changes in amounts of morphogen production, and are consistent with the idea that release is regulated.

## Hh gradients form independently in the wing disc, ASP, and myoblasts

To characterize how signaling proteins are apportioned among the cells they target, we analyzed Hh signaling in a uniquely positioned group of Hh-responding cells near the Hh-producing cells of the wing disc notum primordium. Cells in this region include cells of the wing disc A compartment, the ASP and myoblasts, mesenchymal cells that cover most of the disc A compartment in this region and extend over a portion of the P compartment as well. These myoblasts will develop into the flight muscles in the adult. Because of the close proximity of these cells to Hh-producing cells in the disc, and because no other Hh-producing cells are as close, we presume that the Hh they receive originates from this one source (*Hatori and Kornberg, 2020*). We designated an area 150 µm×150 µm

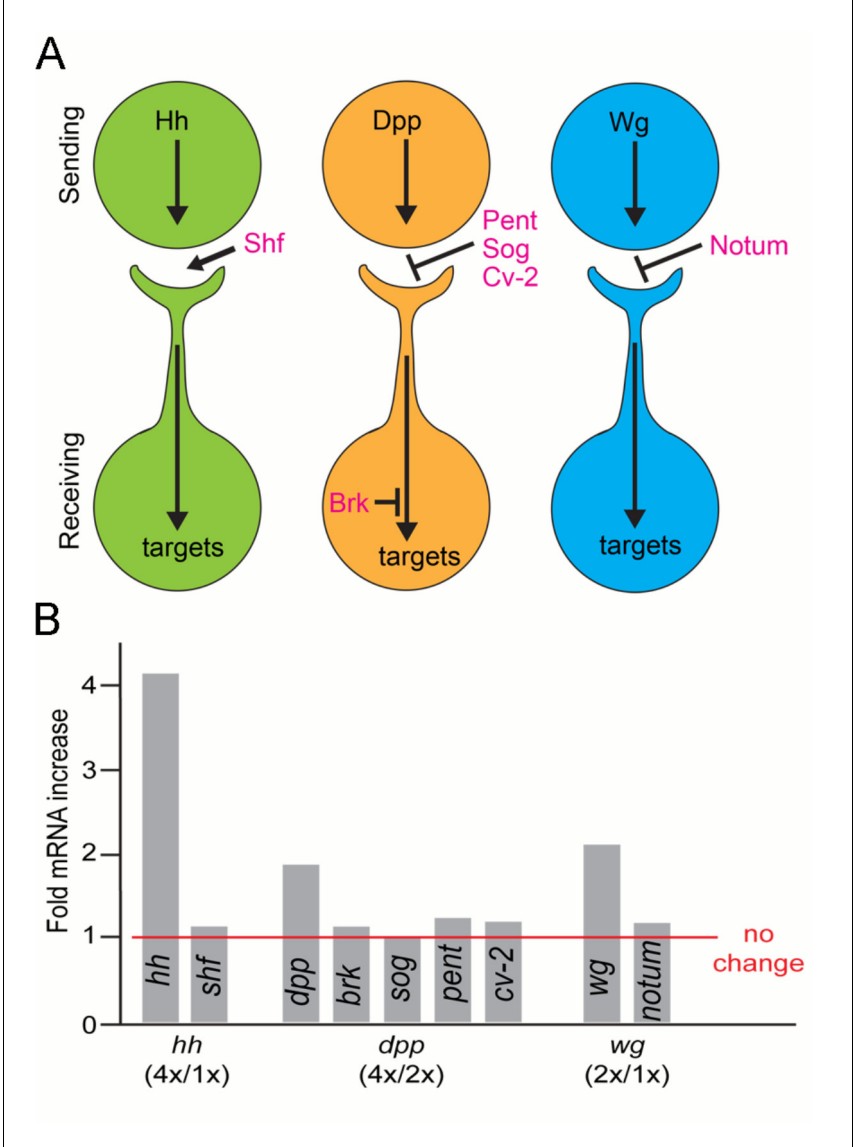

**Figure 6.** Expression of morphogen signaling modulators is not affected by varying amounts of morphogen production. (**A**) Schematic showing where morphogen signaling modulators are predicted to function in the context of cytoneme-mediated exchange. Shf, an extracellular factor that facilitates Hh dispersion; Pent, Sog, and Cv-2, extracellular inhibitors of Dpp signaling; Brk, a transcriptional repressor of Dpp signal transduction; Notum, an extracellular inhibitor of Wg signaling. (**B**) Bar graph showing the levels of morphogen signaling modulator mRNA as determined by qPCR. Bars represent the ratio between the change in mRNA levels relative to predicted RNA increase that scales with gene copy.

The online version of this article includes the following source data for figure 6:

**Source data 1.** Relative mRNA amounts.

that includes all the Hh-responding cells in the notum, ASP, and myoblasts, as a Hh 'microenvironment' (*Figure 7A–C*).

We investigated the behavior of Hh in the three cell populations of this microenvironment, testing if signaling in one tissue is influenced by the amount of Hh the others take up. The experiment distinguishes whether regulated Hh export creates a common pool of signaling protein that is shared among target cells, or if export is independently directed to target cells. If uptake is from a common pool, reducing the number of target cells is predicted to increase uptake and signaling in the target cells that remain. Experiments that increased amounts of extracellular and diffusible morphogens

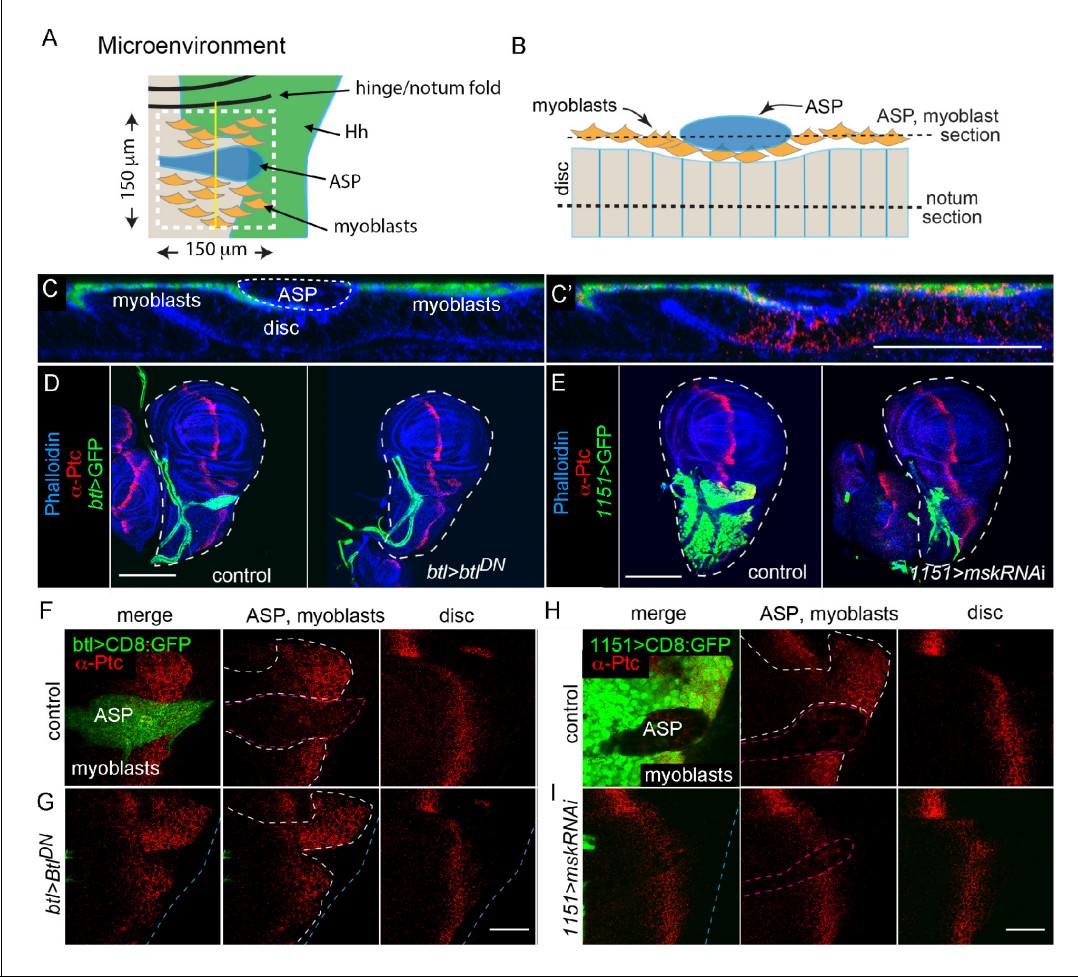

**Figure 7.** Hh distributions in the ASP, myoblast, and the notum primordium are not inter-dependent. (A) Schematic showing the microenvironment (white dashed line) in the notum primordium with myoblasts (orange), ASP (blue), Hh-expressing notum cells (green), and notum A compartment (beige). (B) Schematic showing cross-section of the microenvironment at the yellow line in (A). (C) Confocal image of the cross-section shown in (B); myoblasts (green), phalloidin staining (blue), and α-Ptc staining (red). Scale bar: 50 µm. (D) Wing discs stained with α-Ptc antibody (red) and phalloidin (blue) for control genotype (WT) and (E) ASP ablation genotype (btl-Gal4>Btl$^{DN}$); trachea and ASP marked by CD8:GFP (green) driven by btl-Gal4. Scale bar: 100 µm. (F) α-Ptc staining (red) of the microenvironment for control genotype (WT, ASP marked by CD8:GFP [green] driven by btl-Gal4; outlined by red dashed line in middle panel) and (G) ASP ablation genotype (btl-Gal4>Btl$^{DN}$); white dashed lines surround myoblasts, blue dashed lines indicate the notum primordium. (H, I) Similar to (F, G) but with myoblast ablation; myoblasts marked CD8:GFP (green), ablated by knockdown of msk (1151-Gal4>mskRNAi). Scale bar: 50 µm. Genotypes: (C) 1151-Gal4/+; UAS-CD8:GFP/+; (F, G) control (btl-Gal4 UAS-CD8:GFP/+); btl>Btl$^{DN}$ (btl-Gal4 UAS-CD8:GFP/UAS-Btl$^{DN}$); (H, I) control (1151-Gal4/+; UAS-CD8:GFP/+; 1151>mskRNAi [1151-Gal4/+; UAS-CD8:GFP/UAS-mskRNAi]).

The online version of this article includes the following figure supplement(s) for figure 7:

**Figure supplement 1.** Unchanged size of wing disc with ablation of ASP or myoblasts.

have shown that target cells are capable of responding to increases that are available for uptake. Examples include overexpression of a nonlipidated form of Hh (HhN) in the wing disc and addition of FGF-soaked beads in the chick limb bud; in both experiments, increased signaling in target cells was observed (*Callejo et al., 2006*; *Cohn et al., 1995*).

We genetically ablated the ASP and myoblasts (*Figure 7D,E*), and monitored Ptc expression as a readout of Hh signaling in the remaining tissues (*Figure 7F–I*). The ASP does not develop when tracheal cells overexpress Btl$^{DN}$ (*Du et al., 2018*; *Sato and Kornberg, 2002*), a dominant-negative mutant FGFR protein (*Reichman-Fried and Shilo, 1995*) that inhibits signaling by disc-produced FGF ligand Branchless (Bnl) (*Sato and Kornberg, 2002*). Tracheal overexpression of Cut, a transcription factor that negatively regulates FGF signaling (*Du et al., 2018*; *Pitsouli and Perrimon, 2013*),

also ablates the ASP (*Figure 7—figure supplement 1A*). Neither the presence of Btl[DN] or overexpressed Cut in the tracheal cells had an apparent effect on the growth and morphogenesis of the disc or myoblasts (*Figure 7—figure supplement 1A,B*). In the absence of an ASP (and of the Hh target cells in the ASP), the amounts of Ptc in the disc and myoblasts were indistinguishable from controls (*Figure 7F,G*, *Figure 7—figure supplement 1C*). To ablate myoblasts, we ectopically expressed *moleskin* (*msk*) RNAi. Msk is a nuclear importer of the FGF transcriptional activator ERK (*Vishal et al., 2017*). *mskRNAi* expression in the myoblasts had no apparent effect on the wing disc (*Figure 7—figure supplement 1E*), but reduced the myoblast population and decreased the size of the ASP (*Figure 7—figure supplement 1F*). This ASP phenotype is consistent with our previous findings that ASP growth and morphogenesis are dependent on Notch signaling from the myoblasts (*Huang and Kornberg, 2015*). Whereas the total amount of Ptc in the reduced population of myoblasts decreased under conditions of *msk* expression, Ptc expression in the disc was indistinguishable from controls (*Figure 7H,I*, *Figure 7—figure supplement 1G*).

These experiments and the results are summarized in *Figure 8A–H*. The genotypes we tested eliminate either the ASP or myoblasts (*Figure 8A–F*), and showed that Hh signaling in the disc and myoblasts was not dependent on Hh uptake by the ASP (*Figure 8G*). In addition, despite the fact that the Hh-responding myoblasts in the microenvironment represent an area 20% greater than the area of Hh-responding disc cells at the stage these experiments were conducted, Hh signaling in the disc was not dependent on Hh uptake by myoblasts (*Figure 8H*). We conclude that the Hh target cells in the ASP and myoblasts do not change the delivery of Hh to wing disc cells, and therefore

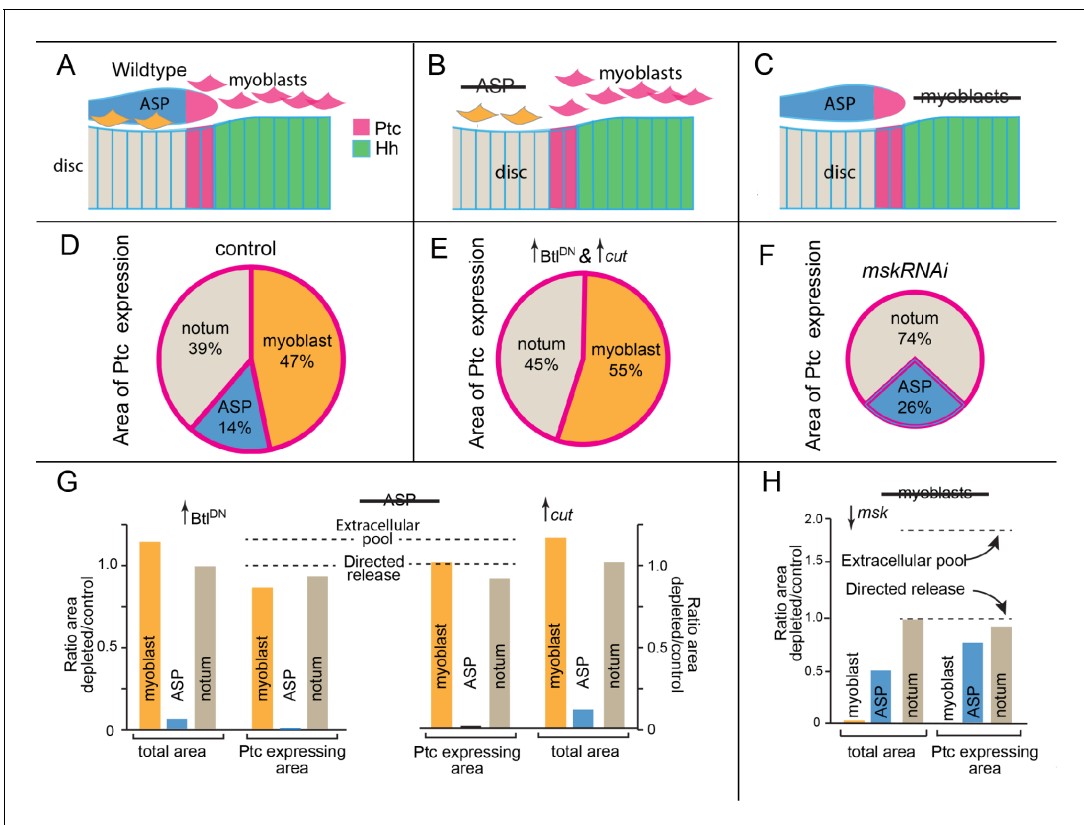

**Figure 8.** Hh signaling in the notum microenvironment. (A–C) Cartoons depicting Hh-expressing (green) and Ptc-expressing cells (pink) in microenvironment of (A) control (wildtype), (B) no ASP (*btl*G4>Btl[DN] and *btl*G4>Cut), and (C) no myoblasts (*1141*G4>mskRNAi). (D–F) Pie graphs depicting the fraction of microenvironment that expresses Ptc in control (D), no ASP (E), and no myoblasts (F) genotypes. (G, H) Bar graphs quantifying total area and Ptc-expressing areas of myoblasts, ASP, notum (total) in control (WT) and ASP-ablation genotype (*btl-Gal4*>Btl[DN]); n=5 and 4 (control and ASP ablation, respectively). Dashed lines indicate predicted changes in Ptc-expressing area under extracellular pool model of dispersion or directed release model of dispersion. (H) Similar to (G) but with the myoblasts depleted by the expression of mskRNAi in the myoblasts (*1141Gal4*>mskRNAi); n=5.

that the spatial patterns of Hh signaling and Hh transport in the disc form independently of the presence or absence of other target cells.

## Discussion

Protein secretion is characterized as either constitutive, such that synthesis and discharge are linked and concurrent, or regulated, such that proteins are made and stored until release is stimulated (*Kelly, 1985*). Antibody-producing lymphocytes are examples of constitutive secretory cells (*Holodick et al., 2010*). Examples of cells that regulate secretion include endocrine pancreatic cells that discharge hormones into the circulatory system, and neurons that pass signals to target cells at chemical synapses. The work reported here demonstrates that delivery of Hh, Dpp, and Wg to target cells is regulated and is not dependent on the amount of protein produced or on constitutive release—that differences in production as much as 4× for Hh, 2× for Dpp, and 7× for Wg did not change the amount of uptake or signal transduction. This work also reports that delivery of Hh to a tissue is not influenced by the complexity or size of the target fields—that the presence or absence of Hh-receiving tracheal cells and myoblasts was of no consequence to Hh signaling in the same region of the wing disc. The conclusion we draw is that the contours of Hh, Dpp, and Wg distributions are determined by controlled exchanges between producing and receiving cells, not by constitutive release from producing cells. This is an important, defining feature of the process that disperses these proteins across tissues.

Diffusion models of morphogen gradient formation are based on the idea that producing cells create a pool of secreted, extracellular signaling protein whose distributions are determined by interactions with extracellular components that non-producing cells contribute—such as receptors, ECM proteins, and negative regulators that absorb or bind passively diffusing protein (*Madamanchi et al., 2021*; *Stapornwongkul and Vincent, 2021*). Although these models assume that morphogen proteins are released constitutively from producing cells, our findings that the amount of signaling protein in a target field is a small fraction of the amount produced and is not proportional to production, do not explicitly invalidate them. Factors contributed by non-producing cells in the signaling domain might, in theory, feedback to compensate for variations in levels of protein in an extracellular pool. However, measures of expression of negative regulators of signal transduction for Hh (*shf*), Dpp (*brk, sog, pent, cv-2*), and Wg (*Notum*) detected no changes in the genotypes we analyzed that had different gene copy numbers (*Figure 6*).

In late third instar larvae, the distal portion of the tracheal ASP commingles with mesenchymal myoblasts on the basal surface of the wing disc anterior to the A/P compartment border (*Figure 1A*). All the ASP, myoblast, and disc cells that are within approximately 60 μm of the Hh-expressing, P compartment disc cells activate Hh signaling (*Hatori and Kornberg, 2020*). Thus, despite the cell cycle, shape, constitution, and fate differences between the cells in this microenvironment, the primary determinant of Hh signaling appears to be distance from producing cells. If Hh were released into the extracellular space within this microenvironment and if the ASP, myoblast, and disc cells shared available Hh, we might expect that the distance over which Hh spreads and the extent of Hh signaling would depend on the number of recipient cells in this Hh target field. It does not, genetic conditions that reduced the number of cells in the target field did not increase the number of remaining cells in the microenvironment that activated Hh signaling (*Figure 8*). This result shows that in the microenvironment, Hh uptake is not determined by either production levels, cell type, or number of other cells that also take up Hh. This finding is not consistent with the idea that Hh populates a shared extracellular pool from which different cells draw, a central tenet of diffusion models. Our interpretation is that this finding is consistent with cytoneme-mediated signaling and the idea that dissemination of Hh involves direct cell-to-cell exchanges between one producing and one receiving cell. Our results suggest that these cell-to-cell exchanges are regulated and are insensitive to modest increases or decreases in Hh production. The ~20× excess in production relative to amounts transferred to target cells may buffer the system and ensure that Hh is not limiting for the process that regulates its transfer. And because the exchanges are restricted to interactions between single pairs of cells, they are insensitive to increases or decreases in the number of other cells in the target field.

The idea that morphogen gradients form by diffusion originated before morphogens were discovered to be signaling proteins, and the first attempt to model gradient formation based on chemical

and physical principles assumed that they were small organic molecules that diffuse freely into and out of cells (*Crick, 1970*). Protein moving in an extracellular environment prior to receptor-mediated uptake has been modeled with more complex mathematics (*Lander, 2007*; *Madamanchi et al., 2021*), but diffusion-based dissemination is still without direct evidence. Studies of morphogen proteins expressed at physiological levels and in normal conditions that have been interpreted as supporting diffusion-based dissemination have used methods that do not distinguish cell-free protein from cell-bound protein. In addition, these studies have not been carried out in conditions in which cytonemes could be imaged (*Stapornwongkul and Vincent, 2021*).

Diffusion-based dissemination has been inferred indirectly from distributions of signaling proteins in normal and mutant contexts. Mutants with defective heparan sulfate proteoglycans that do not distribute morphogens normally are examples, but it has yet to be established whether the observed effects were due to inhibition of protein movement in extracellular space (*Bishop et al., 2007*; *Häcker et al., 2005*; *McGough et al., 2020*; *Mii and Takada, 2020*) or to inhibition of cytoneme function (*Bischoff et al., 2013*; *González-Méndez et al., 2017*). Diffusion-based dissemination has also been inferred from responses to protein released from an implanted bead loaded with protein (*Meyers and Martin, 1999*) or from a micropipette (*de la Torre et al., 1997*), and from the behavior of protein spreading from sites of ectopic overexpression (*Nowak et al., 2011*). However, the protein in these experiments may not disperse by normal routes. In our experiments, Hh release was not gated if Hh was produced at excessively high levels by cells that were genetically engineered for overexpression, in contrast to Hh protein produced at close to normal levels (*Figure 1*). We suggest that protein released from beads, pipets, or overexpressing cells may not reflect the normal state.

Studies that have examined the robustness of morphogen signaling systems to variations in either production or response (*Li et al., 2018*; *Zhang et al., 2020*).

Evidence for cytoneme-mediated transfer of signaling proteins at cell-cell contacts is both genetic and histologic. Cytonemes defective for proteins that provide essential functions to neuronal synapses such as the cell adhesion protein Capricious, the calcium-binding protein Synaptotagmin-4, and potassium rectifying channel Irk-2 do not make normal numbers of functional synaptic contacts, do not disseminate signaling proteins, and are signaling deficient (*Huang et al., 2019*; *Roy et al., 2014*). Hh, Dpp, and Bnl/FGF are visible moving along cytonemes that extend from producing cells and link with receiving cells, and are also visible after uptake colocalized with their respective receptors in cytonemes that extend from receiving cells and link to producing cells (*Capdevila and Guerrero, 1994*; *Du et al., 2018*; *Huang et al., 2019*; *Roy et al., 2014*; *Torroja et al., 2004*). Thus, the evidence for cytoneme-based dissemination is direct and strong.

The process that distributes signaling proteins into concentration gradients that decline with increasing distance from source cells appears to have several components. One regulates the number of cytonemes that link producing and receiving cells. In the wing disc and ASP systems, the number of cytonemes linking producing and receiving cells correlates with signaling strength—cells far from signal sources and low signaling have fewer cytonemes than do cells close to signal sources and high signaling (*Du et al., 2018*; *Roy et al., 2014*). For Branchless/FGF signaling in the ASP, positive feedback that increases the number of relatively short cytonemes of cells that have high signaling levels and are close to source cells, and negative feedback that decreases the number of relatively long cytonemes in cells that have lower signaling levels and are farther from source cells. This system of regulation contributes to the formation of a spatial gradient of cytoneme number and concentration gradient of Branchless/FGF (*Du et al., 2018*). Similar feedback systems may sculpt the cytoneme gradients that disperse Hh and Dpp. Another regulatory mechanism involves feedback responses to signaling. Hh signaling, for example, enhances expression of Ptc, a receptor that binds and sequesters Hh, thereby reducing Hh dispersion and shaping the contour and extent of the Hh gradient (*Chen and Struhl, 1996*; *Li et al., 2018*).

A third regulatory mechanism is reported here—the gating which releases constant amounts of Hh, Dpp, and Wg independently of levels of production. Although we did not identify the rate-limiting step or steps that set these amounts, we made several observations that are relevant. We found that the amount of Hh detected at the basal surface of producing cells was constant and independent of level of production, suggesting that placement at the basal membrane is gated. This idea is consistent with the observation that levels of extracellular Hh were less than total Hh in these cells, but the possibilities remain that release from producing cells and/or uptake by receiving cells may also be regulated. Our investigations of cytoneme biology have been guided by known features of

neurons and neuronal signaling, and we have identified many features of cytoneme-mediated signaling that are analogous. These include spatially-specific signaling at synapses that link cells at distances of <40 nm (*Roy et al., 2014*), synaptic localization of proteins such as the voltage-gated calcium channel and Synaptotagmin, essential roles for the glutamate receptor and glutamate transporter, and trans-synaptic stimulation of calcium transients (*Huang et al., 2019*). And calcium-dependent release of glutamate is essential for both cytoneme-mediated signaling and for glutamatergic excitatory neuronal synapses. It remains for further investigations to determine how morphogen proteins are released from producing cells and are exposed to receiving cells, and if protein transfers at cytoneme synapses are controlled.

# Materials and methods

**Key resources table**

| Reagent type (species) or resource | Designation | Source or reference | Identifiers | Additional information |
|---|---|---|---|---|
| Cell line (*Drosophila*) | *hh^{ac}* | *Lee et al., 1992* | | |
| Cell line (*Drosophila*) | *dpp^{H46}* | *Irish and Gelbart, 1987* | | |
| Cell line (*Drosophila*) | *wg^{RF}* | *Pérez-Garijo et al., 2009* | | |
| Transfected construct | *40k Hh BAC* | *Chen et al., 2017* | | |
| Transfected construct | *40k Hh:GFP BAC* | *Chen et al., 2017* | | |
| Transfected construct | *100k Hh BAC* | *Chen et al., 2017* | | |
| Transfected construct | *Dpp BAC* | this study | | |
| Transfected construct | *Dpp:Cherry* | *Fereres et al., 2019* | | |
| Transfected construct | *wg-Gal4* | *Giráldez et al., 2002* | | |
| Transfected construct | *1151-Gal4* | *Roy and Vijay Raghavan, 1997* | | |
| Transfected construct | *UAS-Wg:GFP* | *Pfeiffer et al., 2002* | | |
| Transfected construct | *UAS-mCD8:GFP* | *Roy et al., 2011* | | |
| Transfected construct | *btl-LHG* | *Roy et al., 2014* | | |
| Transfected construct | *lexO-Cherry: CAAX* | from Konrad Basler | | |
| Transfected construct | *UAS-Cut* | *Hardiman et al., 2002* | | |
| Transfected construct | *UAS-btl^{DN}* | *Reichman-Fried and Shilo, 1995* | | |
| Transfected construct | *UAS-mskRNAi* | Bloomington *Drosophila* Stock Center | #27572 | |
| Antibody | α-Ptc | DSHB, Apa1 | | 1/500 |
| Antibody | α-Hh | from Phillip Ingham | | 1/500 |
| Antibody | α-GFP | Roche | #11814460001 | 1/500; 2/500 for extracellular staining |

*Continued on next page*

*Continued*

| Reagent type (species) or resource | Designation | Source or reference | Identifiers | Additional information |
|---|---|---|---|---|
| Antibody | α-Dpp-Prodomain | *Akiyama and Gibson, 2015* | | 1/500 |
| Antibody | α-RFP | Rockland | #:600-401-379 | 1/500 |
| Antibody | α-Wg | DSHB, 4D4 | | 1/500; 3/500 for extracellular staining |
| Antibody | α-Sens | *Nolo et al., 2000* | | 01/00 |
| Antibody | α-En | DSHB, 4D9 | | 01/25 |
| Antibody | α-Knot | *Crozatier and Vincent, 1999* | | 1/500 |
| Antibody | α-Salm | *Zhang et al., 2011* | | 1/500 |
| Antibody | α-Dll | *McKay et al., 2009* | | 1/500 |
| Antibody | goat α-mouse IgG, Alexa Fluor 488 | Invitrogen | A-11001 | 1/500 |
| Antibody | goat α-mouse IgG, Alexa Fluor 555 | Invitrogen | A-21422 | 1/500 |
| Antibody | goat α-rabbit IgG, Alexa Fluor 488 | Invitrogen | A-11008 | 1/500 |
| Antibody | goat α-rabbit IgG, Alexa Fluor 555 | Invitrogen | A-21428 | 1/500 |
| Antibody | goat α-rat IgG, Alexa Fluor 555 | Invitrogen | A-21434 | 1/500 |
| Antibody | goat α-rat IgG, Alexa Fluor 488 | Invitrogen | A-11006 | 1/500 |
| Other | Vectashield Antifade Mounting Medium | Vector Laboratories | H-1000-10 | |

## Fly culture

Flies were cultured in standard cornmeal and agar medium at 25°C; all crosses were at 25°C, except the expression of *Cut*. To express *Cut* in the ASP, *btl-Gal4/UAS-Cut; Gal80$^{ts}$/+* was incubated at 18°C until early L3 and transferred to 29°C until late L3.

## qPCR analysis of *hh* gene expression

Wing discs were dissected in phosphate-buffered saline (PBS), RNA was extracted using RNeasy Micro Kit (Qiagen), and cDNA was synthesized using the High Capacity RNA-to-cDNA Kit (Applied Biosystems). qPCR was performed with SensiFast Sybr green (Bioline). For each genotype, 3–4 replicates of 5 wing discs were analyzed. Actin was the internal control and fold differences in relative mRNA levels between genotypes were calculated as $2^{-\Delta\Delta Ct}$.

## Immunohistochemistry, fluorescent imaging, and image analysis

Wing discs together with Tr2 trachea were dissected in PBS, fixed in 4% formaldehyde in PBS (25 min), and washed on a rocking rotator in PBS+0.3% Triton X-100 (PBST) (3×10 min), and incubated in Roche blocking solution (1 hr). Primary antibodies were diluted in Roche Blocking solution and incubated with discs overnight at 4°C (12 hr/overnight). Discs were washed in PBST (3×10 min) on a rocking rotator, incubated with secondary antibody diluted in Roche blocking solution (2 hr), washed in PBST (3×10 min), and after removal of PBST, mounted in Vectashield (Vector Labs). All procedures were at room temperature except for primary antibody incubation. The domains of α-Ptc, α-Sense, α-En, and α-pMAD staining were measured in ImageJ from single optical sections at the basolateral part of the wing disc.

## Intensity measurements of proteins

Average intensity quantifications for projections of α-Hh, α-GFP, α-Cherry, and α-Wg staining were calculated for segments of optical sections spanning 20 μm from the most apical side of the wing pouch cells. This segment was chosen because the basal sides of the wing discs are folded and therefore problematic to quantify. Background measurements were taken in equivalent areas distant from staining regions and were subtracted.

The comparison between total and extracellular Hh is a relative estimate because fluors, antibodies, laser intensities, and gain settings were not the same for the two protocols. Wg detected by the extracellular staining protocol in producing and adjacent receiving cells was measured in the

indicated 30 µm×90 µm area, with the producing area defined by a 30 µm×10 µm rectangle and the receiving area defined by two 30 µm×40 µm rectangles. Distance of fluorescence was measured manually. Size differences between discs were small and were not taken into account. All measurements of intensity were with ImageJ.

## Areas of Ptc expression

For *Figure 8G, H, K*, the dotted lines for the predicted extracellular pool were calculated assuming that the total area of Ptc expression is constant and an increase in remaining tissues compensates for the absence of ablated tissue. The ratio for directed release was set at 1.00 based on the assumption that Ptc-expressing area for the remaining tissues would not change under conditions of tissue ablation.

### ASP ablation

Calculated ratio of depleted/control for extracellular pool=1.00/(% of Ptc-expressing area in notum [0.39]+myoblast [0.47] before ablation)=1.16.

### Myoblast ablation

Calculated ratio of depleted/control for extracellular pool=1.00/(% of Ptc-expressing area in notum [0.39]+ASP [0.14])=1.89.

## Quantification of cytoneme density

To observe cytonemes, unfixed preparations were observed using the hanging drop method (*Huang and Kornberg, 2016*). Images were acquired using the FV3000 Olympus Confocal microscope with GaAsP PMT detectors. Images were analyzed and processed with ImageJ and Photoshop.

Maximum intensity projection image of the whole volume of the ASP was used to count the number of cytonemes in the bulb. To calculate the density of cytoneme per µm, the number of cytonemes was divided by the perimeter of the bulb of the ASP.

## Dextran uptake analysis

As adopted from *Torroja et al., 2004*: third Instar larval wing discs were dissected and incubated in 3.7 mM Red dextran (lysine fixable, MW 3000, Molecular Probe) in M3 Media at 25℃ (5 min), washed 5× in ice-cold M3 media (2 min each), fixed in 4% paraformaldehyde/PBS (40 min at 4℃), fixed in 4% PFA/PBS at room temperature (20 min), and washed 2× in PBT (10 min each). α-GFP staining was as above.

## Statistics

Error bars indicate SD and statistical significance was calculated by unpaired Student's t-tests. P value of $<0.05$ is designated as statistically significant.

## Acknowledgements

The authors thank H Bellen, M Gibson, X Lin, R Mann, and M Crozatier-Borde for antibodies and the Bloomington Stock Center and Vienna *Drosophila* Resource Center for fly stocks. This work was funded by NIH T32HL007185 to RH and R35GM122548 to T.B.K.

## Additional information

### Funding

| Funder | Grant reference number | Author |
| --- | --- | --- |
| National Institute of General Medical Sciences | R35GM122548 | Thomas B Kornberg |
| National Institutes of Health | T32HL007185 | Ryo Hatori |

The funders had no role in study design, data collection and interpretation, or the decision to submit the work for publication.

## Author contributions
Ryo Hatori, Conceptualization, Investigation, Methodology, Writing - original draft, Writing - review and editing; Brent M Wood, Guilherme Oliveira Barbosa, Investigation, Writing - review and editing; Thomas B Kornberg, Conceptualization, Supervision, Funding acquisition, Writing - original draft, Project administration, Writing - review and editing

## Author ORCIDs
Ryo Hatori (iD) https://orcid.org/0000-0003-2224-5802
Guilherme Oliveira Barbosa (iD) http://orcid.org/0000-0002-5881-0896
Thomas B Kornberg (iD) https://orcid.org/0000-0002-6879-7066

## Decision letter and Author response
Decision letter https://doi.org/10.7554/eLife.71744.sa1
Author response https://doi.org/10.7554/eLife.71744.sa2

## Additional files

### Supplementary files
• Transparent reporting form

### Data availability
All data generated or analysed during this study are included in the manuscript and supporting files.

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
