## [Decision Letter]

[Editors' note: this paper was reviewed by Review Commons.]

**Acceptance summary:**

This paper tackles the question of whether the distribution of signaling ligands in cells of the wing disc is regulated analogous to neurotransmitters at chemical neuronal synapses. Specifically is the distribution of Hh, Wg, and Dpp dependent and proportional to the amount produced? The data suggest that delivery of these ligands to target cells is regulated in both amount and destination.

---

## [Author Response]

Reviewer #1:1. It has been already published (e.g. Matusek et al 2014) that over-expression of Hh through the UAS/Gal4 system in Hh producing cells is able to increase signalling activity, extending the Hh graded distribution. The authors should discuss it and/or offer an explanation of the resulting increase in cell signalling.

Done; please see revised text and revised Figures 1, 2 New data added to Fig. 1C, D, E, F, G, H.

p5: “We also measured hh RNA in wing discs with 2 WT hh genes and transgenes containing hh-Gal4 and UAS-hh:GFP, and observed that the amount increased approximately 15X over WT.”

2. In the same line of thought it would be interesting to test whether this kind of over-expression in the wing imaginal disc, might change responses in the ASP.

Done; please see revised Figure 5 New data added to Fig. 5B,C.

p12: “In contrast, Hh over-expression driven by hh-Gal4 reduced the stalk and increased the extent of En expression. […] The sensitivity to hh-Gal4 driven overexpression indicates that the capacity of the ASP system to buffer against different levels of expression is limited.”

3. The authors should also present ex vivo staining of morphogens (extracellular levels at least for Hh and Wg), to control for possible effects coming from changes in the processing or degradation rate within producing or receiving cells.

Please see revised Figure 2 for Hh extracellular staining New data added to Fig. 2E,F.

p9: “Gene dosage dependence of extracellular Hh

We applied an extracellular staining protocol that detects antibodies bound to preparations of nonpermeabilized and unfixed cells (Strigini and Cohen, 2000). […] Bulk endocytic uptake by A compartment cells monitored by dextran uptake did not change with hh gene dosage (Fig. 2 Supplement 3).”

Please see revised Figure 4 for Wg extracellular staining. New data added to Fig. 4D.

p11: “We investigated the distribution of Wg using the extracellular staining protocol and detected low levels of basolateral Wg in Wg-producing (Fig, 4, Supplement2). […] This protocol does not distinguish between protein on either the cell surface or associated with cytonemes, but these results suggest that delivery and release to receiving cells from the basolateral membrane is rate-limiting.”

Also Methods:

p30: “The comparison between total and extracellular Hh is a relative estimate because fluors, antibodies, laser intensities, and gain settings were not the same for the two protocols. […] All measurements of intensity were with ImageJ.”

4. To quantify the read out of the Hh, Wg and Dpp signalling gradients the expression of additional targets should be monitored, for instance, for Hh (En, Col, Dpp Ci), for Wg (Dll and Vg) and for Dpp (Brk, Spalt, Omb).

Done; please see Supplemental Figure 1-supplement 3 for Knot, Supplemental Figure 3-supplement 1 for Spalt, and Supplemental Figure 4-supplement 1 for Dll p6: Expression of Knot (Kn), which like Ptc expression is regulated by Hh signaling (Vervoort et al., 1999), did not change in genotypes with 1-4 hh gene copies (Fig. 1 Supplement 3).

p10: “Similarly, expression of Spalt (Salm), which is regulated by Dpp signaling in the wing disc (de Celis et al., 1996), did not change in genotypes with two or four dpp gene copies (Fig. 3 supplement 1).”

p11: “a-Dll antibody detects a region of expression that overlaps Wg-expressing cells (Zecca et al., 1996), and the pattern of Dll expression also was not different in these genotypes (Fig. 4 supplement 1).”

5. Regarding the methods used for image analysis, the authors should state the measuring method used for florescence, is it grey mean value? Was it adjusted for background signalling? Was the section used determined by the maximum width within the apico-basal planes? How was distance of fluorescence signal measured? Was it manually measured? Did it take into account the potential size differences between discs? How was Hh signal quantified?

Done p30: “Average intensity quantifications for projections of α-Hh, α-GFP, α-Cherry, and α-Wg staining were calculated for segments of optical sections spanning 20 from the most apical side of the wing pouch cells. […] Background measurements taken in equivalent areas distant from staining regions and were subtracted.”

6. Statistical analysis is adequate but ideally it would be helpful to include a "power analysis", determining whether the N used is sufficient for a non-significant result. For instance, Ptc patterns in Figure 1 panel D do not look similar in different genotypes.

The images in the Fig 1D panels are single optical sections that were chosen to be representative, and the ones in the revised figure are better representations. The Fig 1E graph quantifies values from optical sections spanning 20 mm from the apical surface and more accurately reflect the Ptc domains. We are not familiar with the suggested “power analysis”.

7. The results shown in Figure 6B are difficult to understand.

Please see revised Figure 6.

8. In the results section, Expression of modulators of morphogen protein signalling, the sentence "This insensitivity to Hh amounts suggests that Shf does not control Hh release".... should be changed as the fact that levels of shf expression are not modified by the doses of Hh does not imply that Shf does not have a role in controlling Hh release.

Text has been revised.

“We quantified shf expression in discs with one and four hh genes by quantifying shf transcripts with qPCR; no change in shf mRNA was detected in these genotypes (Fig. 6A,B). This insensitivity to the tested changes in Hh amounts suggests that Shf does not control Hh release. brinker (brk), pentagone (Pent; aka magu), short gastrulation (sog), and crossveinless-2 (Cv-2) negatively affect Dpp signaling.”

9. Regarding the possibility of cytoneme involvement in the regulated transport/release, can changes in the ASP number of cytonemes be expected by over-expression of Hh in the wing disc, in hhts mutants or in smo- or ptc- clones in the ASP?

Reference to relevant experiments in Hatori and Kornberg (2020) has been added.

10. Would it be possible to regulate the number of cytonemes emanating from the ASP, for instance by increasing FGF signalling, and to look at the Hh signalling response in the ASP?

Revised Figure 5 includes new data showing increased signaling in conditions of Ihog over-expression.

Legend: “(F) Ectopic over-expression of Ihog in the ASP reduced the stalk and increased extent of α-En staining (green). (G) Width of α-Ptc staining band in the wing discs ectopically over-expressing Ihog in the indicated genotypes; differences between 1 and 2 hh copies statistically significant (P<0.05) (Abbreviations as in Fig. 1.)”

Minor comments:1. In Figure 1 panel D, a profile of intensity and a reconstruction of a cross section would be very useful to visualise maximum intensity changes and how levels are distributed in the AP axis.

Figure 1 Supplements 1 and 2 show Ptc cross sections and intensity profiles.

Panel E. Ptc protein alone should not be used to quantify the read out of the Hh signalling gradient, En, Ci, Dpp and Col are also good targets to report it.

Figure 1 Supplement 3 shows results for analyses of Knot expression.

2. In Figure 2, panel A and B, should also show an apico/basal cross section of the wing discs. In addition, it should include, or present instead, quantification of ex vivo staining (extracellular levels), as the total Hh includes unprocessed Hh that cannot reach the plasma membrane to be released.

Figure 2 Supplement 1 shows apico/basal cross section. Figure 2E,F shows results for extracellular staining.

3. In Figure 3 panel E, would be better to show the expression of other Dpp targets such as Spalt, Brinker or Omb.

Figure 3 Supplement 1 shows results for analyses of Spalt expression.

4. In Figure 4 to better analyse the effect of GFP dilution, flies expressing endogenous wg-GFP are actually available, and extracellular staining of Wg in the basolateral part of the producing cells would show if the rate-limiting step occurs before or after release. In panel C, C' monitoring for low threshold targets such as Dll or Vg expression after changing Wg doses would be more adequate.

Figure 4 Supplement 1 shows results for Dll expression.

5. Through the text, it would be better to use Hh-GFP rather than green coloured letters to avoid confusion.

Corrected in text except for Legends.

6. In page 7 first paragraph, a sentence is duplicated "Hh amounts in anterior compartments are not detectably different with 1, 2, 3, or 4 hh genes".

Corrected.

7. Page 9 Figure 4 B’ is not mentioned within the results section Wingless production and signalling in the wing disc.

Corrected.

p11: “Wg antibody staining showed that Wg production is proportional to gene copy number in discs with one and two wg genes, and that the wg-Gal4 driver generated approximately seven times more Wg than a single endogenous gene (Fig. 4A,A’,B). Despite the differences in expression between these genotypes, the amount of Wg in the neighboring cells that received Wg was unchanged (Fig. 4A,A’,B).”

8. Page 11 at the end of the second paragraph, the sentence "We monitored Hh signalling in this region by Ptc expression and determined that..." is incomplete.

Corrected.

p13: “We monitored Ptc expression at the wing disc A/P compartment border in genotypes with 1, 2 or 3 hh genes and observed that the width of the Ptc stripe in discs with two copies increased approximately 35% relative to discs with one (ANOVA and Tukey p<0.05), but no significant difference between discs with two or three copies (Fig. 5G).”

9. In Figure 3, it should be written: Bar graph quantifying α-Cherry antibody staining in sending (I) and receiving (I') regions for indicated genotypes.

Corrected.

p24: “Bar graphs quantifying α-Cherry antibody staining in sending (I) and receiving regions (I’) for indicated genotypes; error bars indicate SD;”

10. In the legend of Figure 3, the panels M and J are referred but they are not actually in the figure.

Corrected.

11. In Figure 5, panels A and B are inverted according to figure legend.

Corrected.

12. In Figure 7, panel E the figure legend indicates: yellow dashed lines mark the compartment boundary, blue arrows mark the extent of myoblast Ptc expression, green arrows mark extent of notum Ptc expression. Left column: ASP, myoblast section in (B).... However, there are no yellow dashed lines, neither blue or green arrows on actual figure.

Corrected.

13. For Figure 7, panel G, the legend should include an explanation of how the total areas and Ptc extension distances were measured.

Corrected.

For Figure 8 (G,H,K), the dotted lines for the predicted extracellular pool was calculated assuming that total area of Ptc expression is constant and increase in remaining tissues compensates for the absence of ablated tissue. The ratio for directed release was set at 1.00 based on the assumption that Ptc expressing area for the remaining tissues would not change under conditions of tissue ablation.

ASP ablation: Calculated ratio of depleted/control for extracellular pool = 1.00/(% of Ptc expressing area in notum (0.39) + myoblast (0.47) before ablation) = 1.16

Myoblast ablation: Calculated ratio of depleted/control for extracellular pool = 1.00/(% of Ptc expressing area in notum (0.39) + ASP (0.14)) = 1.89

14. In figure 7, panel I, the legend indicates that orange arrows mark extent of Ptc expression in the ASP however no orange arrows are shown in panel.

Corrected.

References:1. The paper by Zhang, Zhao et al., 2020 in which the authors propose that the wing disc well tolerate Hh production changes should be cited and discussed.

Done. p7: “Previous studies characterized the robustness of Hh signaling to variations in either production or response, assuming that release from producing cells is constitutive and that robustness is solely an attribute of the signal transduction process (Li et al., 2018; Zhang et al., 2020). We consider two possible alternatives.”

2. The Stanganello et al 2015 reference, demonstrates that Wnt signalling is cytoneme dependent but not for Drosophila Wg.

Corrected p4: “In the wing disc, transport of Dpp is cytoneme-mediated (Huang and Kornberg, 2015; Roy et al., 2014) and although the role of cytonemes in Wg dispersion have not been investigated, Wnt signaling in zebrafish is cytoneme-mediated (Stanganello et al., 2015, p. 2016).”

3. Also, there are two recent papers on cytoneme mediated Hh signalling that proposed for Hh a synaptic like process that are not referenced in the introduction (González-Méndez et al., 2017, 2010).

Corrected p3: “Specialized filopodia called cytonemes are conduits that transport and transfer signaling proteins to target cells at synaptic contacts (González-Méndez et al., 2020, 2017; Kornberg, 2016).”

Reviewer #2 (Evidence, reproducibility and clarity (Required)):Ryo Hatori and Thomas Kornberg´s manuscript utilized the Drosophila wing as model system in a very elegant manner (elegant experiments, elegant presentation and elegant description of results) to present evidence that morphogen release (Hh, Dpp and Wg) from signaling centers is regulated; in other words, that the amount of morphogen that target cells receive is constant and independent of the amount of morphogen that is produced by the source (by comparing experimental settings where the number of copies of the morphogen-encoding gene is either halved, normal or increased). Authors also present evidence that changes in the number of target tissues does not alter the amount of morphogen that is received by the other tissues. In general, I do not have major comments on the manuscript. I can only congratulate authors for this very nice paper. However, I would like to suggest two major changes.First, the discussion is too long, widespread and some paragraphs do not fit in the paper. Thus, I would suggest authors to improve it. Second, I would suggest authors to propose alternative models to explain their results, besides the cytoneme-driven model. I am sure authors might be able to do.

The Discussion has been revised to clarify its logic and reasoning. Although more complex alternative models that involve multiple different mechanisms of dispersion could be proposed (our data shows that signaling is cytoneme-dependent but does not rule out the existence of free, non-cytoneme associated morphogen), we try in the limited space available to explain the reasoning for the cytoneme model. We are not sure what alternatives the reviewer is seeking.

Minor changes include explaining what ASP or myoblasts are in the results section, so that the general reader can understand the experimental settings,

Text revised. p12: “We first analyzed Hh signaling and cytoneme densities in the air sac primordium (ASP), a tracheal branch which is physically attached to the wing disc and which extends cytonemes to the disc that mediate the uptake of Hh, Dpp, and Bnl (Chen et al., 2017a; Du et al., 2018; Hatori and Kornberg, 2020; Roy et al., 2014).”

p14: “To characterize how signaling proteins are apportioned among the cells they target, we analyzed Hh signaling in a uniquely positioned group of Hh-responding cells near the Hh-producing cells of the wing disc notum primordium. […] These myoblasts will develop into the flight muscles in the adult.”

Rephrase some sentences (e.g. "The importance of regulation by morphogen gradients to growth, cell fate and patterning underlies the imperative to understand how morphogens disperse across tissues") and be stricter about which reference(s) to be included through the text (try to be honest with the first observers of the experimental observations that are being reviewed/described).

We thought we had referenced appropriately and would appreciate suggestions where referencing is deficient.

I believe this manuscript is a very strong candidate for developmental biology-oriented journal.Minor comments:Last para of page 6: Fig 2A and not Fig 1A is the one that presents the results.

Corrected.

Last para of pag 11: there is an editing problem so that one sentence is not complete.

Corrected.

Materials and methods: dilution of Abs, code nrs of commercial Abs should be included. References to academic Abs are lacking. Fly stocks nomenclature to be revised. Statistics section necessary.

Statistics section added; new Table with reagent data added p28: **“**Fly Lines and Antibodies”.

p31: “Statistics

Error bars indicate standard deviation (SD) and statistical significance was calculated by unpaired student’s t-tests. P value of <0.05 is designated as statistically significant.”

Reviewer #3 (Evidence, reproducibility and clarity (Required)):The paper "Regulated delivery controls Drosophila Hedgehog, Wingless and Decapentaplegic signaling" from Ryo Hatori and Thomas B. Kornberg describes the very interesting finding that the amount of produced morphogens does not alter the signaling strength of morphogen signaling. The experiments and hypothesis are nicely done and highly relevant, as it is a long standing and controversially discussed question.By gene dosage of Hh, Dpp and Wg the authors can show that although protein levels increase in the producing cells, the amount of morphogens in target the signaling and the functional effects are not altered. It is a common understanding, that the imaginal wing disc is a robust system and can compensate for different manipulations in generating a functional wing. Yet, to see this demonstrate in experiments and especially the titration experiments with fluorescently tagged morphogens is really great.My main point of criticism is concerning the lack of addressing endosomal trafficking in producing cells as a lever to the extracellular pool of morphogens:Page 7 "These results are consistent with the idea that most Hh produced in the posterior compartment is not released (and does not signal), and that Hh export is not linked directly to production." The authors argue that if the produced amount does not change the signaling strength than regulated delivery has to be the determining factor. As the number of cytonemes of receiving cells is unchanged, the authors should also consider that the rate of endocytosis in producing cells could change upon increasing production of morphogens by recycling and uptake assays. I would like to see the extracellular level of the morphogens and whether they change with gene dosage. What if producing cells, by morphogen signaling themselves, sense the relevant amount of morphogens that need to be presented on the surface?

We agree and are also intrigued by the possibility that producing cells communicate with receiving cells in the process of cytoneme-mediated transfer, but would prefer not to speculate about such a process in this manuscript.

The role of endocytosis in producing cells in the regulation of morphogen signaling is a controversial subject for itself with some recent publications available: D’Angelo Dev Cell 2018, Hemalatha PNAS, 2016 Munthe et al, J Cell Sci 2020, Linnemannstöns et al, Development 2020, Witte et al., Development 2020.

Please see Figure 2 Supplement 2 with new data showing that dextran uptake is not sensitive to Hh production.

p9: Bulk endocytic uptake by A compartment cells monitored by dextran uptake did not change with hh gene dosage (Fig. 2 Supplement3).

Some aspects of the manuscript can be improved to make it better:In the discussion, I feel that two different ideas are mixed, comparison with the neuronal model (constitutive versus regulated release) and the question how morphogen delivery is modulated if not by the amounts produced. What if cytonemes and other forms of morphogens spreading coexist and are the delivery forms for different target cells. In that respect, the discussion is a bit to one-sided. This should be more precisely discussed in the light of the last figure.

The Discussion has been revised to clarify its logic and reasoning. Although more complex alternative models that involve multiple different mechanisms of dispersion could be proposed, data from our studies and the studies of others show that signaling to every known target cell for every signaling protein that has been investigated is cytoneme-dependent. These findings do not rule out the existence of free, non-cytoneme associated morphogen, but show that signaling is cytoneme-dependent.

The last figure is not well enough described, because the set-up is more complex, more details are needed to follow through even more people from the wing disc field.

Corrected.

Please see revised Figures 7 and 8.

There are some unfinished sentences and double sentences in the manuscript. For example: page 7 " that Hh amounts in anterior compartments are not detectably different with 1, 2, 3, or 4 hh genes (Fig. 2A,B).," Page 14 "...and are possibilities" .

Corrected.

Page 9 "monitored ASP cytonemes" give a short explanation for ASP for a broader audience Page 10 As the cytonemes stem from target cells, that should be stated again in the results part, again to make it clearer for a broader audience.

Done; see above.

Page 14 “are sequestered in intracellular vesicles prior to release (Callejo et al., 2011; Gradilla et al., 2018;Yamazaki et al., 2016)” consider more endocytosis papers.

Not relevant to revised Discussion.